# MetaGPT: Meta Programming for a Multi-Agent Collaborative Framework

**Sirui Hong**[1,*] **Mingchen Zhuge**[2*]**, Jonathan Chen**[1]**, Xiawu Zheng**[3]**, Yuheng Cheng**[4]**,
Ceyao Zhang**[4]**, Jinlin Wang**[1]**, Zili Wang, Steven Ka Shing Yau**[5]**, Zijuan Lin**[4]**,
Liyang Zhou**[6]**, Chenyu Ran**[1]**, Lingfeng Xiao**[1,7]**, Chenglin Wu**[1,†] **Jürgen Schmidhuber**[2,8]

[1]DeepWisdom, [2]AI Initiative, King Abdullah University of Science and Technology,
[3]Xiamen University,     [4]The Chinese University of Hong Kong, Shenzhen,
[5]Nanjing University,     [6]University of Pennsylvania,
[7]University of California, Berkeley,     [8]The Swiss AI Lab IDSIA/USI/SUPSI

## Abstract

Remarkable progress has been made on automated problem solving through societies of agents based on large language models (LLMs). Existing LLM-based multi-agent systems can already solve simple dialogue tasks. Solutions to more complex tasks, however, are complicated through logic inconsistencies due to cascading hallucinations caused by naively chaining LLMs. Here we introduce MetaGPT, an innovative meta-programming framework incorporating efficient human workflows into LLM-based multi-agent collaborations. MetaGPT encodes Standardized Operating Procedures (SOPs) into prompt sequences for more streamlined workflows, thus allowing agents with human-like domain expertise to verify intermediate results and reduce errors. MetaGPT utilizes an assembly line paradigm to assign diverse roles to various agents, efficiently breaking down complex tasks into subtasks involving many agents working together. On collaborative software engineering benchmarks, MetaGPT generates more coherent solutions than previous chat-based multi-agent systems. Our project can be found at https://github.com/geekan/MetaGPT.

## 1 Introduction

Autonomous agents utilizing Large Language Models (LLMs) offer promising opportunities to enhance and replicate human workflows. In real-world applications, however, existing systems (Park et al., 2023; Zhuge et al., 2023; Cai et al., 2023; Wang et al., 2023c; Li et al., 2023; Du et al., 2023; Liang et al., 2023; Hao et al., 2023) tend to oversimplify the complexities. They struggle to achieve effective, coherent, and accurate problem-solving processes, particularly when there is a need for meaningful collaborative interaction (Chen et al., 2024; Zhang et al., 2023; Dong et al., 2023; Zhou et al., 2023; Qian et al., 2023).

Through extensive collaborative practice, humans have developed widely accepted Standardized Operating Procedures (SOPs) across various domains (Belbin, 2012; Manifesto, 2001; DeMarco & Lister, 2013). These SOPs play a critical role in supporting task decomposition and effective coordination. Furthermore, SOPs outline the responsibilities of each team member, while establishing standards for intermediate outputs. Well-defined SOPs improve the consistent and accurate execution of tasks that align with defined roles and quality standards (Belbin, 2012; Manifesto, 2001; DeMarco & Lister, 2013; Wooldridge & Jennings, 1998). For instance, in a software company, Product Managers analyze competition and user needs to create Product Requirements Documents (PRDs) using a standardized structure, to guide the developmental process.

Inspired by such ideas, we design a promising GPT-based Meta-Programming framework called MetaGPT that significantly benefits from SOPs. Unlike other works (Li et al., 2023; Qian et al., 2023), MetaGPT requires agents to generate structured outputs, such as high-quality requirements

---

*These authors contributed equally to this work.

†Chenglin Wu (alexanderwu@fuzhi.ai) is the corresponding author, affiliated with DeepWisdom.

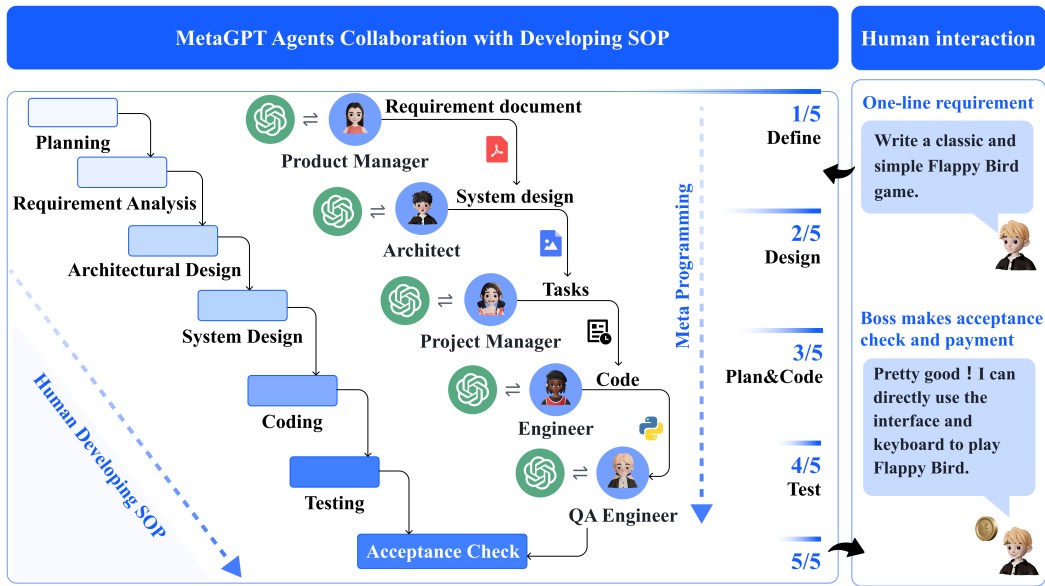

Figure 1: **The software development SOPs between MetaGPT and real-world human teams.** In software engineering, SOPs promote collaboration among various roles. MetaGPT showcases its ability to decompose complex tasks into specific actionable procedures assigned to various roles (e.g., Product Manager, Architect, Engineer, etc.).

documents, design artifacts, flowcharts, and interface specifications. The use of intermediate structured outputs significantly increases the success rate of target code generation. Because it helps maintain consistency in communication, minimizing ambiguities and errors during collaboration. More graphically, in a company simulated by MetaGPT, all employees follow a strict and streamlined workflow, and all their handovers must comply with certain established standards. This reduces the risk of hallucinations caused by idle chatter between LLMs, particularly in role-playing frameworks, like: "*Hi, hello and how are you?*" – Alice (Product Manager); "*Great! Have you had lunch?*" – Bob (Architect).

Benefiting from SOPs, MetaGPT offers a promising approach to meta-programming. In this context, we adopt meta-programming[1] as "programming to program", in contrast to the broader fields of meta learning and "learning to learn" (Schmidhuber, 1987; 1993a; Hochreiter et al., 2001; Schmidhuber, 2006; Finn et al., 2017).

This notion of meta-programming also encompasses earlier efforts like CodeBERT (Feng et al., 2020) and recent projects such as CodeLlama (Rozière et al., 2023) and WizardCoder (Luo et al., 2023). However, MetaGPT stands out as a unique solution that allows for efficient meta-programming through a well-organized group of specialized agents. Each agent has a specific role and expertise, following some established standards. This allows for automatic requirement analysis, system design, code generation, modification, execution, and debugging during runtime, highlighting how agent-based techniques can enhance meta-programming.

To validate the design of MetaGPT, we use publicly available HumanEval (Chen et al., 2021a) and MBPP (Austin et al., 2021) for evaluations. Notably, in code generation benchmarks, MetaGPT achieves a new state-of-the-art (SoTA) with 85.9% and 87.7% in Pass@1. When compared to other popular frameworks for creating complex software projects, such as AutoGPT (Torantulino et al., 2023), LangChain (Chase, 2022), AgentVerse (Chen et al., 2023), and ChatDev (Qian et al., 2023). MetaGPT also stands out in handling higher levels of software complexity and offering extensive functionality. Remarkably, in our experimental evaluations, MetaGPT achieves a $100\%$ task completion rate, demonstrating the robustness and efficiency (time and token costs) of our design.

We summarize our contributions as follows:

---

[1]https://en.wikipedia.org/w/index.php?title=Metaprogramming

• We introduce MetaGPT, a meta-programming framework for multi-agent collaboration based on LLMs. It is highly convenient and flexible, with well-defined functions like role definition and message sharing, making it a useful platform for developing LLM-based multi-agent systems.

• Our innovative integration of human-like SOPs throughout MetaGPT's design significantly enhances its robustness, reducing unproductive collaboration among LLM-based agents. Furthermore, we introduce a novel executive feedback mechanism that debugs and executes code during runtime, significantly elevating code generation quality (e.g., 5.4% absolute improvement on MBPP).

• We achieve state-of-the-art performance on HumanEval (Chen et al., 2021a) and MBPP (Austin et al., 2021). Extensive results convincingly validate MetaGPT, suggesting that it is a promising meta-programming framework for developing LLM-based multi-agent systems.

## 2 RELATED WORK

**Automatic Programming**  The roots of automatic programming reach back deep into the previous century. In 1969, Waldinger & Lee (1969) introduced "PROW," a system designed to accept program specifications written in predicate calculus, generate algorithms, and create LISP implementations (McCarthy, 1978). Balzer (1985) and Soloway (1986) made efforts to advance automatic programming and identified potential methods to achieve it. Recent approaches use natural language processing (NLP) techniques (Ni et al., 2023; Skreta et al., 2023; Feng et al., 2020; Li et al., 2022; Chen et al., 2018; 2021b; Zhang et al., 2023). Automatic programming has grown into an industry delivering paid functions such as Microsoft Copilot. Lately, LLMs-based agents (Yao et al., 2022; Shinn et al., 2023; Lin et al., 2023) have advanced automatic programming development. Among them, ReAct (Yao et al., 2022) and Reflexion (Shinn et al., 2023) utilize a chain of thought prompts (Wei et al., 2022) to generate reasoning trajectories and action plans with LLMs. Both works demonstrate the effectiveness of the ReAct style loop of reasoning as a design paradigm for empowering automatic programming. Additionally, ToolFormer (Schick et al., 2023) can learn how to use external tools through simple APIs. The research most closely aligned with our work by Li et al. (2023) proposes a straightforward role-play framework for programming that involves communication between agents playing different roles. Qian et al. (2023) utilizes multiple agents for software development. Although existing papers (Li et al., 2023; Qian et al., 2023) have improved productivity, they have not fully tapped into effective workflows with structured output formats. This makes it harder to deal with complex software engineering issues.

**LLM-Based Multi-Agent Frameworks**  Recently, LLM-based autonomous agents have gained tremendous interest in both industry and academia (Wang et al., 2023b). Many works (Chen et al., 2024; Wang et al., 2023c; Du et al., 2023; Zhuge et al., 2023; Hao et al., 2023; Akata et al., 2023) have improved the problem-solving abilities of LLMs by integrating discussions among multiple agents. Stable-Alignment (Liu et al., 2023) creates instruction datasets by deriving consensus on value judgments through interactions across a sandbox with LLM agents. Other works focus on sociological phenomena. For example, Generative Agents (Park et al., 2023) creates a "town" of 25 agents to study language interaction, social understanding, and collective memory. In the Natural Language-Based Society of Mind (NLSOM) (Zhuge et al., 2023), agents with different functions interact to solve complex tasks through multiple rounds of "mindstorms." Cai et al. (2023) propose a model for cost reduction by combining large models as tool makers and small models as tool users.

Some works emphasize cooperation and competition related to planning and strategy (Bakhtin et al., 2022); others propose LLM-based economies (Zhuge et al., 2023). These works focus on open-world human behavior simulation, while MetaGPT aims to introduce human practice into multi-agents frameworks. Besides, LLM-based agents face the challenges of "assistant repeated instruction" or "infinite loop of message" (Talebirad & Nadiri, 2023; Li et al., 2023). These challenges become more urgent in task-oriented collaborations, which require consistent and mutually beneficial interactions (Elazar et al., 2021; Wang et al., 2022; Jiang et al., 2023). This motivates our focus on applying advanced concepts such as Standard Operating Procedures in software development to multi-agent frameworks.

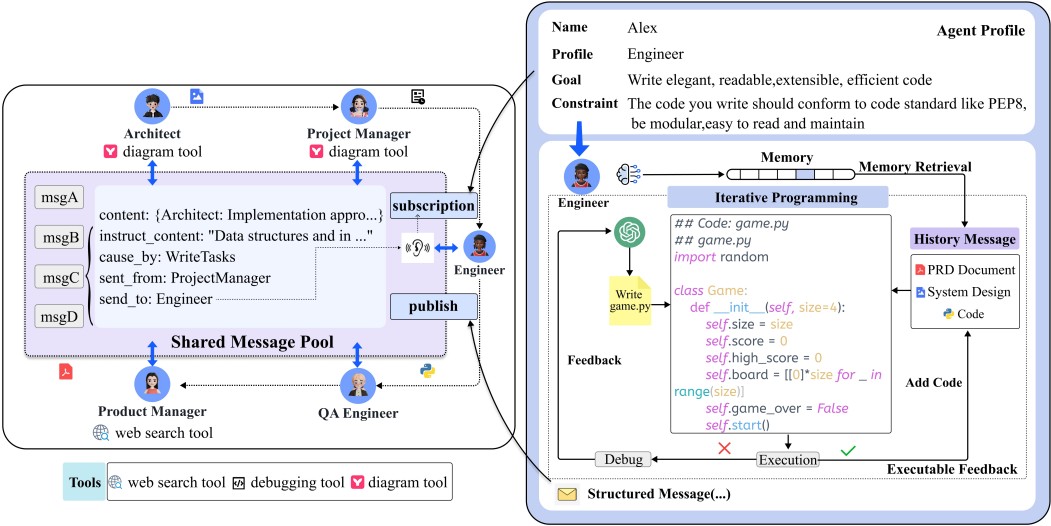

Figure 2: An example of the communication protocol (left) and iterative programming with executable feedback (right). **Left**: Agents use a shared message pool to publish structured messages. They can also subscribe to relevant messages based on their profiles. **Right**: After generating the initial code, the Engineer agent runs and checks for errors. If errors occur, the agent checks past messages stored in memory and compares them with the PRD, system design, and code files.

# 3 METAGPT: A META-PROGRAMMING FRAMEWORK

MetaGPT is a meta-programming framework for LLM-based multi-agent systems. Sec. 3.1 provides an explanation of role specialization, workflow and structured communication in this framework, and illustrates how to organize a multi-agent system within the context of SOPs. Sec. 3.2 presents a communication protocol that enhances role communication efficiency. We also implement structured communication interfaces and an effective publish-subscribe mechanism. These methods enable agents to obtain directional information from other roles and public information from the environment. Finally, we introduce executable feedback—a self-correction mechanism for further enhancing code generation quality during run-time in Sec. 3.3.

## 3.1 AGENTS IN STANDARD OPERATING PROCEDURES

**Specialization of Roles**  Unambiguous role specialization enables the breakdown of complex work into smaller and more specific tasks. Solving complex tasks or problems often requires the collaboration of agents with diverse skills and expertise, each contributing specialized outputs tailored to specific issues.

In a software company, a Product Manager typically conducts business-oriented analysis and derives insights, while a software engineer is responsible for programming. We define five roles in our software company: Product Manager, Architect, Project Manager, Engineer, and QA Engineer, as shown in Figure 1. In MetaGPT, we specify the agent's profile, which includes their name, profile, goal, and constraints for each role. We also initialize the specific context and skills for each role. For instance, a Product Manager can use web search tools, while an Engineer can execute code, as shown in Figure 2. All agents adhere to the React-style behavior as described in Yao et al. (2022).

Every agent monitors the environment (*i.e.*, the message pool in MetaGPT) to spot important observations (*e.g.,*, messages from other agents). These messages can either directly trigger actions or assist in finishing the job.

**Workflow across Agents**  By defining the agents' roles and operational skills, we can establish basic workflows. In our work, we follow SOP in software development, which enables all agents to work in a sequential manner.

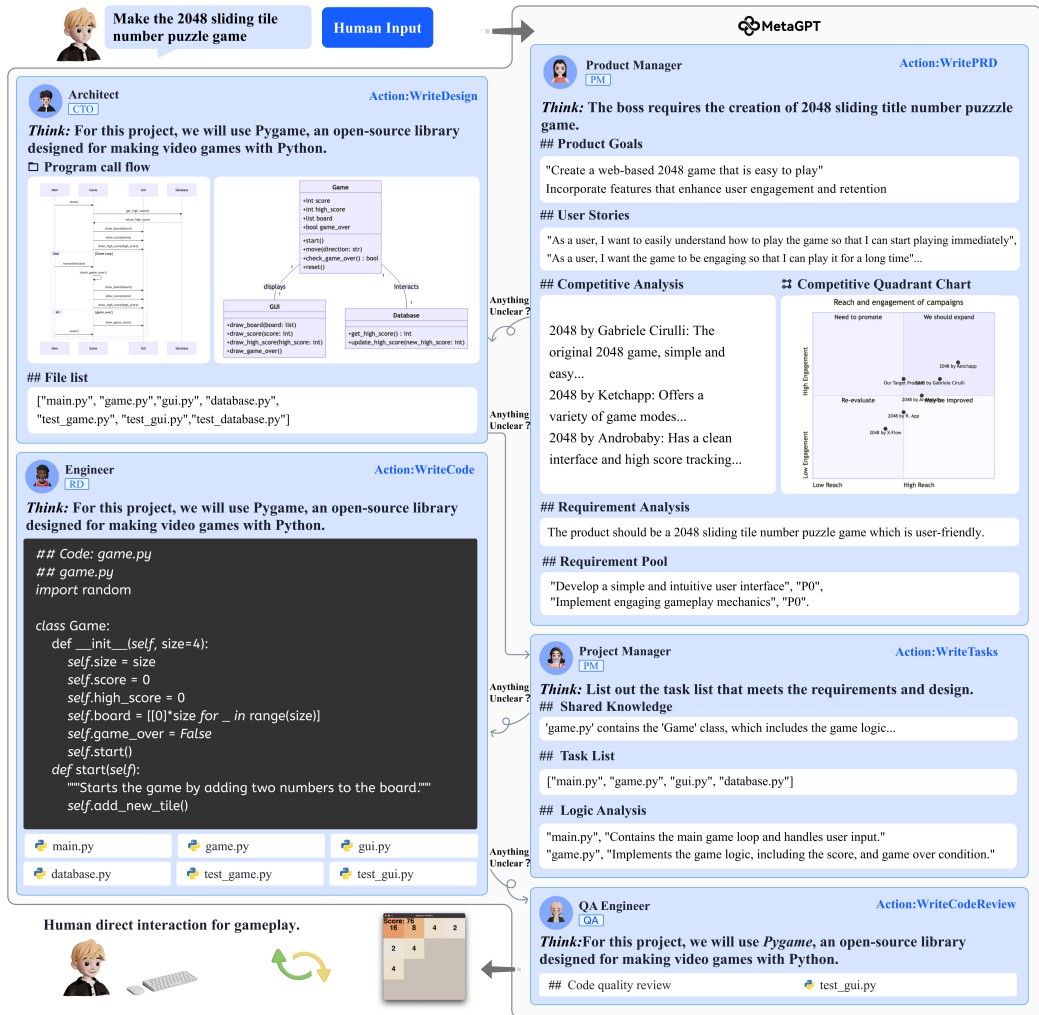

Figure 3: A diagram showing the software development process in MetaGPT, emphasizing its significant dependence on SOPs. The more detailed demonstration can be found in Appendix B.

Specifically, as shown in Figure 1, upon obtaining user requirements, the Product Manager undertakes a thorough analysis, formulating a detailed PRD that includes User Stories and Requirement Pool. This serves as a preliminary functional breakdown. The structured PRD is then passed to the Architect, who translates the requirements into system design components, such as File Lists, Data Structures, and Interface Definitions. Once captured in the system design, the information is directed towards the Project Manager for task distribution. Engineers proceed to execute the designated classes and functions as outlined (detailed in Figure 2). In the following stage, the QA Engineer formulates test cases to enforce stringent code quality. In the final step, MetaGPT produces a meticulously crafted software solution. We provide a detailed schematic (Figure 3) and a concrete instance (Appendix B) of the SOP workflow in MetaGPT.

## 3.2 COMMUNICATION PROTOCOL

**Structured Communication Interfaces**    Most current LLM-based multi-agent frameworks (Li et al., 2023; Zhuge et al., 2023; Zhang et al., 2023; Park et al., 2023) utilize unconstrained natural language as a communication interface. However, despite the versatility of natural language, a question arises: does pure natural language communication suffice for solving complex tasks? For example, in the telephone game (or Chinese

whispers)[2], after several rounds of communication, the original information may be quite distorted. Inspired by human social structures, we propose using structured communication to formulate the communication of agents. We establish a schema and format for each role and request that individuals provide the necessary outputs based on their specific role and context.

As shown in Figure 3, the Architect agent generates two outputs: the system interface design and a sequence flow diagram. These contain system module design and interaction sequences, which serve as important deliverables for Engineers. Unlike ChatDev (Zhao et al., 2023), agents in MetaGPT communicate through documents and diagrams (structured outputs) rather than dialogue. These documents contain all necessary information, preventing irrelevant or missing content.

**Publish-Subscribe Mechanism**  Sharing information is critical in collaboration. For instance, Architects and Engineers often need to reference PRDs. However, communicating this information each time in a one-to-one manner, as indicated by previous work (Li et al., 2023; Zhao et al., 2023; Zhang et al., 2023), can complicate the communication topology, resulting in inefficiencies.

To address this challenge, a viable approach is to store information in a global *message pool*. As shown in Figure 2 (left), we introduce a shared message pool that allows all agents to exchange messages directly. These agents not only *publish* their structured messages in the pool but also access messages from other entities transparently. Any agent can directly retrieve required information from the shared pool, eliminating the need to inquire about other agents and await their responses. This enhances communication efficiency.

Sharing all information with every agent can lead to information overload. During task execution, an agent typically prefers to receive only task-related information and avoid distractions through irrelevant details. Effective management and dissemination of this information play a crucial role. We offer a simple and effective solution-*subscription mechanism* (in Figure 2 (left)). Instead of relying on dialogue, agents utilize role-specific interests to extract relevant information. They can select information to follow based on their role profiles. In practical implementations, an agent activates its action only after receiving all its prerequisite dependencies. As illustrated in Figure 3, the Architect mainly focuses on PRDs provided by the Product Manager, while documents from roles such as the QA Engineer might be of lesser concern.

## 3.3 ITERATIVE PROGRAMMING WITH EXECUTABLE FEEDBACK

In daily programming tasks, the processes of debugging and optimization play important roles. However, existing methods often lack a self-correction mechanism, which leads to unsuccessful code generation. Previous work introduced non-executable code review and self-reflection (Zhao et al., 2023; Yao et al., 2022; Shinn et al., 2023; Dong et al., 2023). However, they still face challenges in ensuring code executability and runtime correctness.

Our first MetaGPT implementations overlooked certain errors during the review process, due to LLM hallucinations (Manakul et al., 2023). To overcome this, after initial code generation, we introduce an executable feedback mechanism to improve the code iteratively. More specifically, as shown in Figure 2, the Engineer is asked to write code based on the original product requirements and design.

This enables the Engineer to continuously improve code using its own historical execution and debugging memory. To obtain additional information, the Engineer writes and executes the corresponding unit test cases, and subsequently receives the test results. If satisfactory, additional development tasks are initiated. Otherwise the Engineer debugs the code before resuming programming. This iterative testing process continues until the test is passed or a maximum of 3 retries is reached.

## 4 EXPERIMENTS

### 4.1 EXPERIMENTAL SETTING

**Datasets**  We use two public benchmarks, HumanEval (Chen et al., 2021a) and MBPP (Austin et al., 2021), and a self-generated, more challenging software development benchmark named Soft-

---

[2]https://en.wikipedia.org/wiki/Chinese_whispers

wareDev: (1) HumanEval includes 164 handwritten programming tasks. These tasks encompass function specifications, descriptions, reference codes, and tests. (2) MBPP consists of 427 Python tasks. These tasks cover core concepts and standard library features and include descriptions, reference codes, and automated tests. (3) Our SoftwareDev dataset is a collection of 70 representative examples of software development tasks, each with its own task prompt (see Table 8). These tasks have diverse scopes (See Figure 5), such as mini-games, image processing algorithms, data visualization. They offer a robust testbed for authentic development tasks. Contrary to previous datasets (Chen et al., 2021a; Austin et al., 2021), SoftwareDev focuses on the engineering aspects. In the comparisons, we randomly select seven representative tasks for evaluation.

**Evaluation Metrics** For HuamnEval and MBPP, we follow the unbiased version of $\text{Pass}@k$ as presented by (Chen et al., 2021a; Dong et al., 2023), to evaluate the functional accuracy of the top-k generated codes: $\text{Pass}@k = \mathbb{E}_{\text{Problems}}\left[1 - \frac{\binom{n-c}{k}}{\binom{n}{k}}\right]$.

For SoftwareDev, we prioritize practical use and evaluate performance through human evaluations (A, E) or statistical analysis (B, C, D): **(A)** Executability: this metric rates code from 1 (failure/non-functional) to 4 (flawless). '1' is for non-functional, '2' for runnable but imperfect, '3' for nearly perfect, and '4' for flawless code. **(B)** Cost: the cost evaluations here include the (1) running time, (2) token usage, and (3) expenses. **(C)** Code Statistics: this includes (1) code files, (2) lines of code per file, and (3) total code lines. **(D)** Productivity: basically, it is defined as the number of token usage divided by the number of lines of code, which refers to the consumption of tokens per code line. **(E)** Human Revision Cost: refers to times of manual code corrections, which tackle problems like package import errors, incorrect class names, or incomplete reference paths. Typically, each correction involves up to 3 lines of code.

**Baselines** We compare our method with recent domain-specific LLMs in the code generation field, including AlphaCode (Li et al., 2022), Incoder (Fried et al., 2022), CodeGeeX (Zheng et al., 2023), CodeGen (Nijkamp et al., 2023), CodeX (Chen et al., 2021a), and CodeT (Chen et al., 2022) and general domain LLMs such as PaLM (Chowdhery et al., 2022), and GPT-4 (OpenAI, 2023). Several results of baselines (such as Incoder, CodeGeeX) are provided by Dong et al. (2023). In HumanEval and MBPP, we slightly modified the prompts to align with response format requirements. These modifications aim to address format-specific issues (i.e., Python problems). With the SoftwareDev benchmark, we provide a comprehensive comparison between MetaGPT, AutoGPT (Torantulino et al., 2023), LangChain (Chase, 2022) with Python Read-Eval-Print Loop (REPL) tool[3], Agent-Verse (Chen et al., 2023), and ChatDev (Qian et al., 2023).

## 4.2 MAIN RESULT

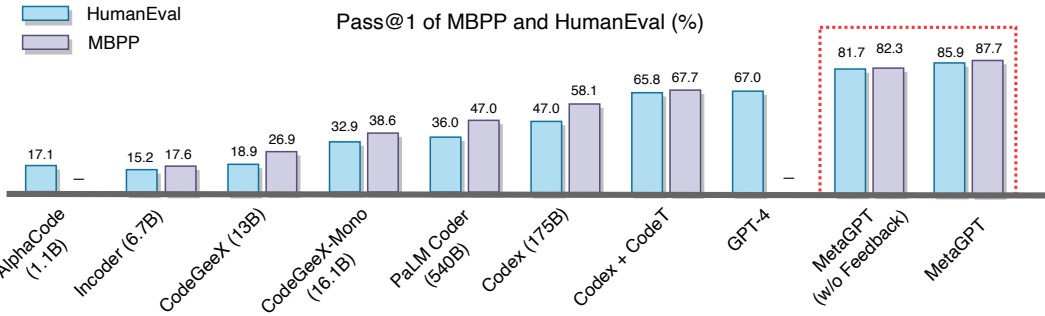

Figure 4: Pass rates on the MBPP and HumanEval with a single attempt.

**Performance** Figure 4 demonstrates that MetaGPT outperforms all preceding approaches in both HumanEval and MBPP benchmarks. When MetaGPT collaborates with GPT-4, it significantly improves the $\text{Pass}@k$ in the HumanEval benchmark compared to GPT-4. It achieves 85.9% and 87.7%

---

[3]https://en.wikipedia.org/wiki/Read–eval–print_loop

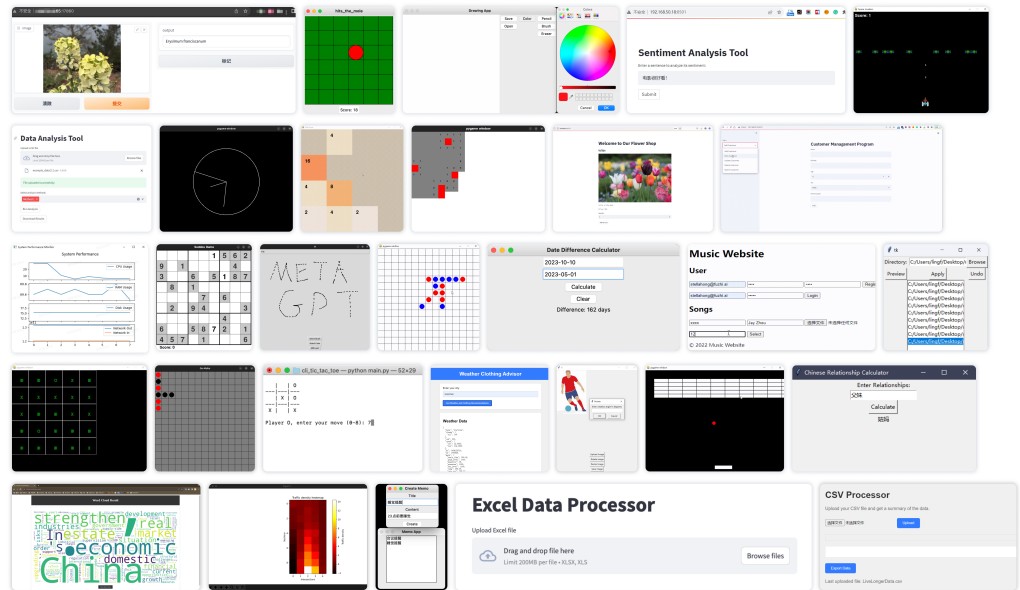

Figure 5: Demo softwares developed by MetaGPT.

in these two public benchmarks. Moreover, as shown in Table 1, MetaGPT outperforms ChatDev on the challenging SoftwareDev dataset in nearly all metrics. For example, considering the executability, MetaGPT achieves a score of 3.75, which is very close to 4 (flawless). Besides, it takes less time (503 seconds), clearly less than ChatDev. Considering the code statistic and the cost of human revision, it also significantly outperforms ChatDev. Although MetaGPT requires more tokens (24,613 or 31,255 compared to 19,292), it needs only 126.5/124.3 tokens to generate one line of code. In contrast, ChatDev uses 248.9 tokens. These results highlight the benefits of SOPs in collaborations between multiple agents. Additionally, we demonstrate the autonomous software generation capabilities of MetaGPT through visualization samples (Figure 5). For additional experiments and analysis, please refer to Appendix C.

Table 1: The statistical analysis on SoftwareDev.

| Statistical Index | ChatDev | MetaGPT w/o Feedback | MetaGPT |
|---|---|---|---|
| **(A)** Executability | 2.25 | 3.67 | **3.75** |
| **(B)** Cost#1: Running Times (s) | 762 | **503** | 541 |
| **(B)** Cost#2: Token Usage | **19,292** | 24,613 | 31,255 |
| **(C)** Code Statistic#1: Code Files | 1.9 | 4.6 | **5.1** |
| **(C)** Code Statistic#2: Lines of Code per File | 40.8 | 42.3 | **49.3** |
| **(C)** Code Statistic#3: Total Code Lines | 77.5 | 194.6 | **251.4** |
| **(D)** Productivity | 248.9 | 126.5 | **124.3** |
| **(E)** Human Revision Cost | 2.5 | 2.25 | **0.83** |

## 4.3 CAPABILITIES ANALYSIS

Compared to open-source baseline methods such as AutoGPT and autonomous agents such as AgentVerse and ChatDev, MetaGPT offers functions for software engineering tasks. As presented in Table 2, our framework encompasses a wide range of abilities to handle complex and specialized development tasks efficiently. Incorporating SOPs (e.g., role-play expertise, structured communication, streamlined workflow) can significantly improve code generation. Other baseline methods

Table 2: **Comparison of capabilities for MetaGPT and other approaches.** '✓' indicates the presence of a specific feature in the corresponding framework, '✗' its absence.

| Framework Capabiliy | AutoGPT | LangChain | AgentVerse | ChatDev | MetaGPT |
|---|---|---|---|---|---|
| PRD generation | ✗ | ✗ | ✗ | ✗ | ✓ |
| Tenical design genenration | ✗ | ✗ | ✗ | ✗ | ✓ |
| API interface generation | ✗ | ✗ | ✗ | ✗ | ✓ |
| Code generation | ✓ | ✓ | ✓ | ✓ | ✓ |
| Precompilation execution | ✗ | ✗ | ✗ | ✗ | ✓ |
| Role-based task management | ✗ | ✗ | ✗ | ✓ | ✓ |
| Code review | ✗ | ✗ | ✓ | ✓ | ✓ |

Table 3: **Ablation study on roles.** '#' denotes 'The number of', 'Product' denotes 'Product manager', and 'Project' denotes 'Project manager'. '✓' indicates the addition of a specific role. 'Revisions' refers to 'Human Revision Cost'.

| Engineer | Product | Architect | Project | #Agents | #Lines | Expense | Revisions | Executability |
|---|---|---|---|---|---|---|---|---|
| ✓ | ✗ | ✗ | ✗ | 1 | 83.0 | **$ 0.915** | 10 | 1.0 |
| ✓ | ✓ | ✗ | ✗ | 2 | 112.0 | $ 1.059 | 6.5 | 2.0 |
| ✓ | ✓ | ✓ | ✗ | 3 | 143.0 | $ 1.204 | 4.0 | 2.5 |
| ✓ | ✓ | ✗ | ✓ | 3 | 205.0 | $ 1.251 | 3.5 | 2.0 |
| ✓ | ✓ | ✓ | ✓ | **4** | **191.0** | $ 1.385 | **2.5** | **4.0** |

can easily integrate SOP-like designs to improve their performance, similar to injecting chain-of-thought (Wei et al., 2022) in LLMs.

### 4.4 ABLATION STUDY

**The Effectiveness of Roles**  To understand the impact of different roles on the final results, we perform two tasks that involve generating effective code and calculating average statistics. When we exclude certain roles, unworkable codes are generated. As indicated by Table 3, the addition of roles different from just the Engineer consistently improves both revisions and executability. While more roles slightly increase the expenses, the overall performance improves noticeably, demonstrating the effectiveness of the various roles.

**The Effectiveness of Executable Feedback Mechanism**  As shown in Figure 4, adding executable feedback into MetaGPT leads to a significant improvement of 4.2% and 5.4% in Pass@1 on HumanEval and MBPP, respectively. Besides, Table 1 shows that the feedback mechanism improves feasibility (3.67 to 3.75) and reduces the cost of human revisions (2.25 to 0.83). These results illustrate how our designed feedback mechanism can produce higher-quality code. Additional quantitative results of MetaGPT and MetaGPT without executable feedback are shown in  Table 4 and Table 9.

### 5 CONCLUSION

This work introduces MetaGPT, a novel meta-programming framework that leverages SOPs to enhance the problem-solving capabilities of multi-agent systems based on Large Language Models (LLMs). MetaGPT models a group of agents as a simulated software company, analogous to simulated towns (Park et al., 2023) and the Minecraft Sandbox in Voyager (Wang et al., 2023a). MetaGPT leverages role specialization, workflow management, and efficient sharing mechanisms such as message pools and subscriptions, rendering it a flexible and portable platform for autonomous agents and multi-agent frameworks. It uses an executable feedback mechanism to enhance code generation quality during runtime. In extensive experiments, MetaGPT achieves state-of-the-art performance on multiple benchmarks. The successful integration of human-like SOPs inspires future research on human-inspired techniques for artificial multi-agent systems. We also view our work as an early attempt to regulate LLM-based multi-agent frameworks. See also the **outlook (Appendix A)**.

**Acknowledgement**

We thank Sarah Salhi, the Executive Secretary of KAUST AI Initiative, and Yuhui Wang, Postdoctoral Fellow at the KAUST AI Initiative, for helping to polish some of the text. We would like to express our gratitude to Wenyi Wang, a PhD student at the KAUST AI Initiative, for providing comprehensive feedback on the paper and for helping to draft the outlook (Appendix A) with Mingchen. We also thank Zongze Xu, the vice president of DeepWisdom, for providing illustrative materials for AgentStore.

**Author Contributions**

Sirui Hong conducted most of the experiments and designed the executable feedback module. She also led the initial version of the write-up, supported by Ceyao Zhang, and also by Jinlin Wang and Zili Wang. Mingchen Zhuge designed the self-improvement module, discussed additional experiments, and led the current write-up. Jonathan Chen helped with the MBPP experiments, outlined the methods section, and contributed to the current write-up. Xiawu Zheng provided valuable guidance, reviewed and edited the paper. Yuheng Cheng contributed to the evaluation metric design and HumanEval experiments. Steven Ka Shing Yau, Zijuan Lin, Liyang Zhou, Lingfeng Xiao helped with the MBPP experiments and comparisons to open-source baseline methods. Chenyu Ran created most of the illustrative figures. Chenglin Wu is the CEO of DeepWisdom, initiated MetaGPT, made the most significant code contributions to it, and advised this project. Jürgen Schmidhuber, Director of the AI Initiative at KAUST and Scientific Director of IDSIA, advised this project and helped with the write-up.

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

# A  OUTLOOK

## A.1  SELF-IMPROVEMENT MECHANISMS

One limitation of the MetaGPT version in the main text of this paper is that each software project is executed independently. However, through active teamwork, a software development team should learn from the experience gained by developing each project, thus becoming more compatible and successful over time.

This is somewhat related to the idea of recursive self-improvement, first informally proposed in 1965 (Good, 1965), with first concrete implementations since 1987 (Schmidhuber, 1987; 1993b; Schmidhuber et al., 1998), culminating in the concept of mathematically optimal self-referential self-improvers (Schmidhuber, 2003; 2009). Generally speaking, a system should learn from experience in the real world, and meta-learn better learning algorithms from experiences of learning, and meta-meta-learn better meta-learning algorithms from experiences of meta-learning, etc., without any limitations except those of computability and physics.

More recent, somewhat related work leverages the reasoning ability of Large Language Models (LLMs) and recursively improves prompts of LLMs, to improve performance on certain downstream tasks (Fernando et al., 2023; Zelikman et al., 2023), analogous to the adaptive prompt engineer of 2015 (Schmidhuber, 2015) where one neural network learns to generate sequence of queries or prompts for another pre-trained neural network whose answers may help the first network to learn new tasks more quickly.

In our present work, we also explore a self-referential mechanism that recursively modifies the constraint prompts of agents based on information they observe during software development. Our initial implementation works as follows. Prior to each project, every agent in the software company reviews previous feedback and makes necessary adjustments to their constraint prompts. This enables them to continuously learn from past project experiences and enhance the overall multi-agent system by improving each individual in the company. We first establish a *handover feedback* action for each agent. This action is responsible for critically summarizing the information received during the development of previous projects and integrating this information in an updated constraint prompt. The summarized information is stored in *long-term memory* such that it can be inherited by future constraint prompt updates. When initiating a new project, each agent starts with a *react* action. Each agent evaluates the received feedback and summarizes how they can improve in a constraint prompt.

One current limitation is that these summary-based optimizations only modify constraints in the specialization of roles (Sec. 3.1) rather than structured communication interfaces in communication protocols (Sec. 3.2). Future advancements are yet to be explored.

## A.2  MULTI-AGENT ECONOMIES

In real-world teamwork, the interaction processes are often not hardcoded. For example, in a software company, the collaboration SOP may change dynamically.

One implementation of such self-organization is discussed in the paper on a "Natural Language-Based Society of Mind" (NLSOM) (Zhuge et al., 2023), which introduced the idea of an "Economy of Minds" (EOM), a Reinforcement Learning (RL) framework for societies of LLMs and other agents. Instead of using standard RL techniques to optimize the total reward of the system through modifications of neural network parameters, EOMs use the principles of supply and demand in free markets to assign credit (money) to those agents that contribute to economic success (reward).

The recent agent-based platform of DeepWisdom (AgentStore[4]) is compatible with the credit assignment concept of EOMs. Each agent in AgentStore provides a list of services with corresponding costs. A convenient API is provided so that human users or agents in the platform can easily purchase services from other agents to accomplish their services. Figure 6 displays the User Interface (UI) of AgentStore, where various agents with different skills are showcased. Besides, individual developers can participate in building new agents and enable collaborative development within the community. Specifically, AgentStore allows users to subscribe to agents according to their demands

---

[4]http://beta.deepwisdom.ai

and pay according to their usage. Moreover, users can purchase additional capabilities to expand the plug-and-play functions of their existing agents. This allows users to gradually upgrade their agents. Within the MetaGPT framework, AgentStore can support the collaboration of various agents. Users can collect several agents together to carry out more complex tasks or projects, and all the agents share and comply with development and communication protocols defined in MetaGPT.

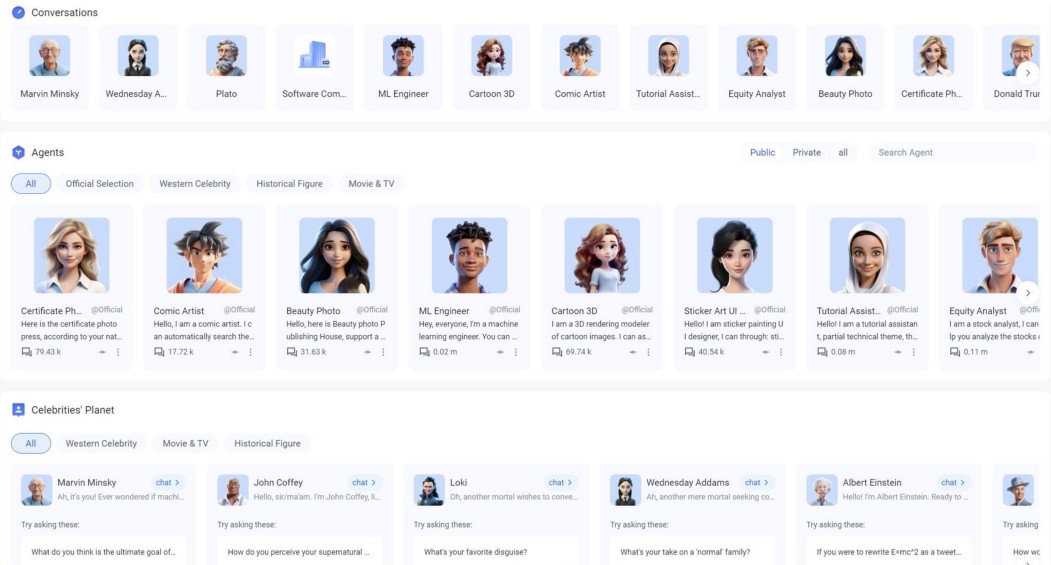

Figure 6: AgentStore is a platform dedicated to serving users in the creation and development of agents within the MetaGPT framework. This platform provides users with an operational interface, allowing users to easily manage a variety of agents with different emotions, personalities, and capabilities for specific tasks.

# B  A DEMO OF THE EXECUTION

In this section, we outline the complete process of software development using MetaGPT. It begins with a user's input command (as shown in Appendix B.1) and ends with software designed according to the user's specifications.

## B.1  USER INPUT

Upon receiving an instruction from the user, MetaGPT collaborates with a professional development team to fulfill the task. Here is a demo of user input:

> **User input**
>
> ```
> 1  write a python3 GUI app such that you can draw an image with it
> ```

## B.2  METAGPT DEVELOPMENT PROCESS

Now we provide a step-by-step explanation of the standardized output process for each agent.

**Product Manager**  The Product Manager generates a Product Requirement Document (PRD), as detailed in the specified documentation. This document encompasses goals, user stories, competitive analysis, requirement analysis and requirement pool. Additionally, a competitive quadrant chart is produced (see Figure 7). Subsequently, these documents and charts are handed over to the architect for system design.

> **Product Requirement Document**
>
> ```
>  1  ## Original Requirements
>  2  The boss requires a Python3 GUI color meter that outputs the RGB values when it moves.
>  3
>  4  ## Product Goals
>  5  ```python
>  6  [
>  7      "Create a user-friendly GUI color meter",
>  8      "Ensure the color meter accurately outputs RGB values",
>  9      "Ensure the color meter updates RGB values in real-time as it moves"
> 10  ]
> 11  ```
> 12
> 13  ## User Stories
> 14  ```python
> 15  [
> 16      "As a user, I want to select any color on the screen, so that I can get its RGB values",
> 17      "As a user, I want the RGB values to update in real-time as I move the color meter, so that I
>        ↪ can get accurate color information",
> 18      "As a user, I want a simple and intuitive interface, so that I can easily use the color meter"
> 19  ]
> 20  ```
> 21
> 22  ## Competitive Analysis
> 23  ```python
> 24  [
> 25      "Color Cop: A popular color picker for Windows. It's simple and easy to use, but lacks real-
>        ↪ time RGB value updates",
> 26      "Just Color Picker: Offers real-time color picking and multiple color code formats. However,
>        ↪ its interface is cluttered and not very user-friendly",
> 27      "Instant Eyedropper: A simple color picker that lives in the system tray. It's convenient but
>        ↪ lacks advanced features",
> 28      "ColorZilla: A color picker extension for browsers. It's convenient for web design but not
>        ↪ suitable for desktop applications",
> 29      "ColorPic: Offers color palettes and mixer tools. It's feature-rich but can be overwhelming
>        ↪ for simple tasks"
> 30  ]
> 31  ```
> ```

**Product Requirement Document**

```
1
2  ## Requirement Analysis
3  The product should be a Python3 GUI application that serves as a color meter. It should allow
        ↪ users to select any color on the screen and display the corresponding RGB values. The RGB
        ↪  values should update in real-time as the color meter moves. The interface should be
        ↪ simple and user-friendly.
4
5  ## Requirement Pool
6  ```python
7  [
8      ("Design a user-friendly GUI for the color meter", "P0"),
9      ("Implement color selection functionality", "P0"),
10     ("Display RGB values of the selected color", "P0"),
11     ("Update RGB values in real-time as the color meter moves", "P0"),
12     ("Test the application for accuracy and performance", "P1")
13 ]
14 ```
15
16 ## UI Design draft
17 The UI should be minimalistic and intuitive. It should primarily consist of a color meter, which
        ↪ could be a magnified area of the screen that the user can move around. The RGB values of
        ↪ the currently selected color should be prominently displayed, updating in real-time as
        ↪ the color meter moves. The layout should be clean and uncluttered, with a focus on
        ↪ functionality and ease of use.
18
19 ## Anything UNCLEAR
20 There are no unclear points.
```

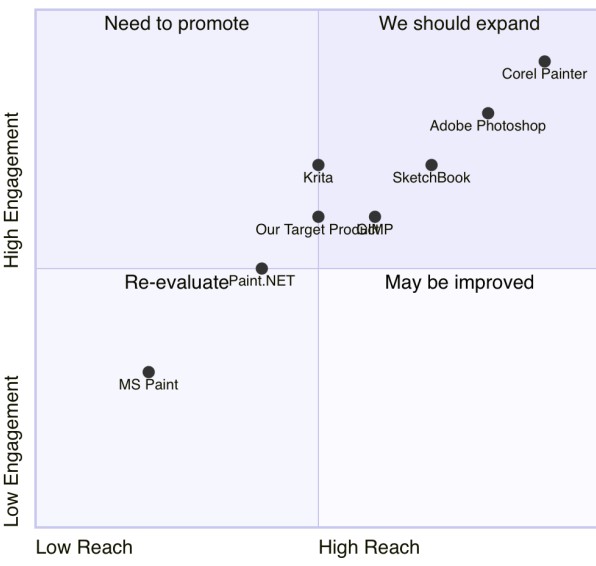

Figure 7: The quadrant chart for the "Drawing App" generated by the *product manager* agent.

**Architect**   Based on the requirements in PRD, the Architect agent devises technical specifications including system architecture diagrams and interface definitions. Initially, the Architect defines the overarching technical trajectory. Subsequently, the project's architecture, including files, classes (Figure 8) and the sequence flow chart (Figure 12), is designed. The Architect's documentation is then given to the project manager for task allocation and execution.

**System design**

```
1  ## Implementation approach
2  We will use Python's Tkinter library to create the GUI for the color meter. Tkinter is a standard
        ↪ Python interface to the Tk GUI toolkit and is included with most Python installations. It
        ↪ is simple to use and powerful enough for our needs.
3
4  The color selection functionality will be implemented using the PIL (Pillow) library, which allows
        ↪ us to capture the screen's pixel data. By moving the mouse, we can get the RGB values of
        ↪ the pixel under the mouse cursor in real-time.
5
6  The RGB values will be displayed on the GUI using Tkinter's Label widget. We will use Python's
        ↪ threading library to update the RGB values in real-time as the color meter moves.
7
8  The application will be tested for accuracy and performance using Python's unittest library.
9
10 ## Python package name
11 ```python
12 "color_meter"
13 ```
14
15 ## File list
16 ```python
17 [
18     "main.py",
19     "color_picker.py",
20     "gui.py",
21     "tests.py"
22 ]
23 ```
24
25 ## Anything UNCLEAR
26 The requirement is clear to me.
```

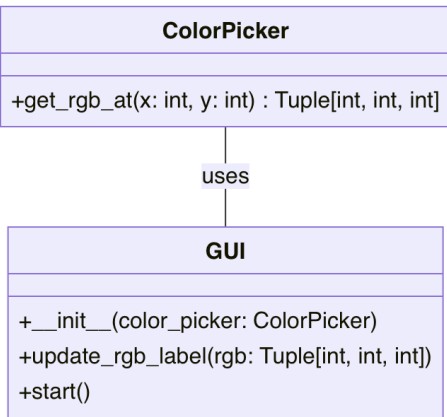

Figure 8: Data structures and interface definitions for the "Drawing App" generated by the *architect* agent.

**Project Manager**   The Project Manager breaks down the project into a task list. Furthermore, each code file is analyzed based on its intended functionality and then treated as a separate task assigned to Engineers.

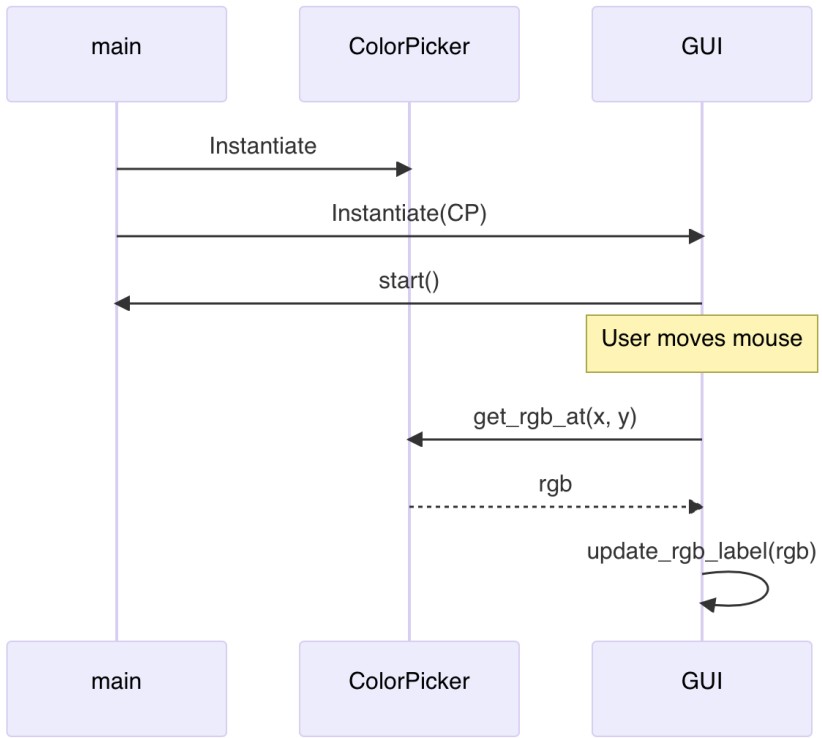

Figure 9: The program call flow for the "Drawing App" generated by the *architect* agent.

**Tasks**

```
1  ## Required Python third-party packages
2
3  ```python
4  """
5  tkinter==8.6
6  Pillow==8.3.1
7  """
8  ```
9
10 ## Required Other language third-party packages
11 ```python
12 """
13 No third-party packages required in other languages.
14 """
15 ```
16
17 ## Full API spec
18 ```python
19 """
20 No APIs are used in this project.
21 """
22 ```
23
24 ## Logic Analysis
25 ```python
26 [
27     ("main.py", "Contains the main application loop and initializes all the other components."),
28     ("canvas.py", "Implements the drawing canvas, responds to mouse events, and uses the selected
       ↪ tool and color."),
29     ("tools.py", "Implements the tool selection interface."),
30     ("color_picker.py", "Implements the color selection interface."),
31     ("file_manager.py", "Implements the file saving and opening functionality.")
32 ]
33 ```
```

**Product Requirement Document**

```
1  ## Task list
2  ```python
3  [
4      "main.py",
5      "canvas.py",
6      "tools.py",
7      "color_picker.py",
8      "file_manager.py"
9  ]
10 ```
11
12 ## Shared Knowledge
13 ```python
14 """
15 The 'Tkinter' library is used for creating the GUI of the application. It provides various widgets
      ↪  like frames and canvases which are used in this project.
16 The 'Pillow' library is used for handling image files. It is used in the 'file_manager.py' for
      ↪  saving and opening image files.
17 The 'main.py' file is the entry point of the application. It initializes all the other components
      ↪  and starts the application loop.
18 """
19 ```
20
21 ## Anything UNCLEAR
22 The project requirements and implementation approach are clear. However, we need to ensure that
      ↪  all team members are familiar with the Tkinter and Pillow libraries. If not, they may
      ↪  need some time to learn these libraries before starting the development.
```

**Engineer** Given the provided file structure and function definitions, an Engineer agent requires only fundamental development skills to complete the development tasks. Due to the large number of files, we present only one auto-generated code file here.

**Generated code (color_picker.py)**

```python
import tkinter as tk
from tkinter import colorchooser
class ColorPicker:
    def __init__(self, root: tk.Tk, color: str = 'black'):
        self.frame = tk.Frame(root)
        self.color = color

        self.color_button = tk.Button(self.frame, text='Color', command=self.select_color)
        self.color_button.pack(fill='x')

    def select_color(self):
        color = colorchooser.askcolor()[1]
        if color is not None:
            self.color = color

    def pack(self, **kwargs):
        self.frame.pack(**kwargs)

    def get_color(self):
        return self.color
```

**QA Engineer** Upon receiving the code output from the Engineer, the QA Engineer generates unit test code and reviews it to identify and fix any bugs, ensuring high-quality software.

**Generated Unit tests (test.py)**

```python
import unittest
import tkinter as tk
# You might want to move the ColorPicker class to another file, let's say color_picker.py
# from color_picker import ColorPicker

class TestColorPicker(unittest.TestCase):

    def setUp(self):
        self.root = tk.Tk()
        self.cp = ColorPicker(self.root)

    def tearDown(self):
        # This is important to close the tk window after tests
        self.root.destroy()

    def test_initial_color(self):
        # Test the default color during initialization
        self.assertEqual(self.cp.get_color(), 'black')

    def test_set_and_get_color(self):
        # Test setting a new color and retrieving it
        new_color = '#ffffff'  # white color
        self.cp.color = new_color
        self.assertEqual(self.cp.get_color(), new_color)

if __name__ == '__main__':
    unittest.main()
```

**Output** Ultimately, as shown in Figure 10, MetaGPT generates a functional application named "Drawing App".

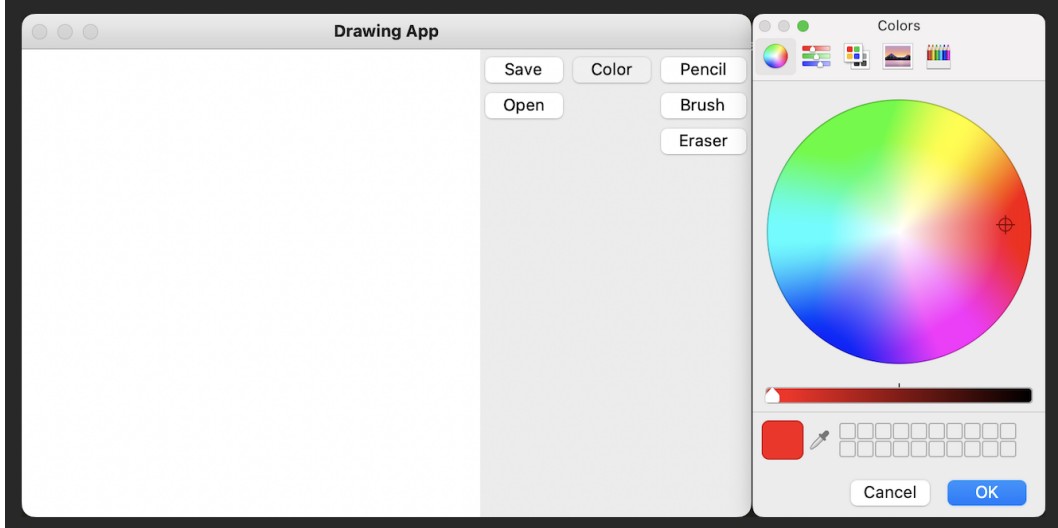

Figure 10: The "Drawing App" generated by MetaGPT.

## C  EXPERIMENTS

### C.1  DETAILS OF THE SOFTWAREDEV DATASET

The SoftwareDev dataset includes 70 diverse software development tasks. Table 8 displays the names and detailed prompts of 11 tasks within the dataset. Note that the first seven tasks listed are used in the main experiments of this paper.

### C.2  ADDITIONAL RESULTS

**Quantitative results of MetaGPT**   As shown in Table 4, MetaGPT achieves an average score of 3.9, surpassing ChatDev's score of 2.1 Zhao et al. (2023), which is based on the Chat chain. Compare the scores of general intelligent algorithms, including AutoGPT Torantulino et al. (2023), which all score 1.0, failing to generate executable code. We observe that the generated code is often short, lacks comprehensive logic, and tends to fail to handle cross-file dependencies correctly.

While models such as AutoGPT (Torantulino et al., 2023), Langchain (Chase, 2022), and Agent-Verse (Chen et al., 2023) display robust general problem-solving capabilities, they lack an essential element for developing complex systems: systematically deconstructing requirements. Conversely, MetaGPT simplifies the process of transforming abstract requirements into detailed class and function designs through a specialized division of labor and SOPs workflow. When compared to Chat-Dev (Zhao et al., 2023), MetaGPT's structured messaging and feedback mechanisms not only reduce loss of communication information but also improve the execution of code.

**Quantitative results of MetaGPT w/o executable feedback**   Table 9 presents the performance of MetaGPT with GPT-4 32K on 11 tasks within the SoftwareDev dataset. It also shows the average performance across all 70 tasks (in the last line). Note that the version of MetaGPT used here is the basic version without the executable feedback mechanism.

**Quantitative results of MetaGPT with different LLMs**   To verify the performance of MetaGPT on different LLMs, we randomly selected 5 SoftwareDev tasks and conducted experiments using GPT-3.5 and Deepseek Coder 33B[5] as backends. As shown in Table 5, the results indicate that although MetaGPT can complete tasks with these LLMs, using GPT-4 as the backend yields superior performance.

---

[5]https://deepseekcoder.github.io

Table 4: **Executability comparison.** The executability scores are on a grading system ranging from '1' to '4'. A score of '1' signifies complete failure, '2' denotes executable code, '3' represents largely satisfying expected workflow, and '4' indicates a perfect match with expectations.

| Task | AutoGPT | LangChain | AgentVerse | ChatDev | MetaGPT |
|---|---|---|---|---|---|
| Flappy bird | 1 | 1 | 1 | 2 | **3** |
| Tank battle game | 1 | 1 | 1 | 2 | **4** |
| 2048 game | 1 | 1 | 1 | 1 | **4** |
| Snake game | 1 | 1 | 1 | 3 | **4** |
| Brick breaker game | 1 | 1 | 1 | 1 | **4** |
| Excel data process | 1 | 1 | 1 | **4** | **4** |
| CRUD manage | 1 | 1 | 1 | 2 | **4** |
| Average score | 1.0 | 1.0 | 1.0 | 2.1 | **3.9** |

Table 5: **Performance of MetaGPT on SoftwareDev using different LLMs as agent backends.**

| Model | Open source | Time(/s) | # Lines | Executability | Revisions |
|---|---|---|---|---|---|
| MetaGPT (w/ GPT-3.5) | ✗ | 75.18 | 161.6 | 2.8 | 2.4 |
| MetaGPT (w/ GPT-4) | ✗ | 552.94 | 178.2 | 3.8 | 1.2 |
| MetaGPT (w/ Deepseek Coder 33B) | ✔ | 1186.20 | 120.2 | 1.4 | 2.6 |

**Impact of Instruction Levels (High-level *v.s.* Detailed Instructions)**  Does the variation in the level of initial input from humans significantly influence performance outcomes? For examples:

1. **High-level prompt**: Create a brick breaker game.
2. **Detailed prompt**: Creating a brick breaker game. In a brick breaker game, the player typically controls a paddle at the bottom of the screen to bounce a ball towards a wall of bricks. The goal is to break all the bricks by hitting them with the ball.

Additional experiments were conducted to investigate this aspect: we selected 5 tasks from SoftwareDev, and constructed detailed prompts for them. Here are the experimental results:

Table 6: **Impact of Instruction Levels.** The executability is scored on a grading system ranging from '1' to '4'. A score of '1' signifies complete failure, '2' denotes runnable code, '3' represents largely expected workflow, and '4' indicates a perfect match to expectations.

| Model | # Word | Time(/s) | Token usage | # Lines | Executability | Productivity | Reversions |
|---|---|---|---|---|---|---|---|
| High-level | 13.2 | 552.9 | 28384.2 | 178.2 | 3.8 | 163.8 | 1.2 |
| Detailed | 42.2 | 567.8 | 29657.0 | 257.0 | 4.0 | 118.0 | 1.6 |

We observe that: detailed prompts lead to better software projects with lower productivity ratios because of clearer requirements and functions, while simple inputs can still generate good enough software using MetaGPT with an executability rating of 3.8, which is comparable to the detailed prompt scenario. (Note that, Productivity = Token usage / Total Code Lines. The lower this ratio, the better.)

**The performance of GPT variants in HumanEval benchmark**  We use the GPT-4's 67% HumanEval score (OpenAI, 2023) as our baseline, acknowledging its acceptance in the HumanEval benchmark. We further extend to experiments(five times) with GPT-4 (gpt-4-0613) and GPT-3.5-Turbo (gpt-3.5-turbo-0613) under various conditions to assess performance. **(A)** We directly called the OpenAI API with the prompt in HumanEval. **(B)** We called the OpenAI API and parsed the code with regex in the response. **(C)** We added an additional system prompt, then called the OpenAI API. The prompt is "You are an AI that only responds with Python code, NOT ENGLISH. You will

be given a function signature and its docstring by the user. Write your full implementation (restate the function signature)." As shown in Table 7, GPT-4 is more sensitive to prompt, code parser, and post-processing results on the HumanEval data set. It is difficult for GPT-3.5-Turbo to return the correct completion code without prompt words.

Table 7: **Performance of GPT models on HumanEval.** Experiments were conducted five times using gpt-4-0613 and gpt-3.5-turbo-0613 with different settings.

| Settings | Model | 1 | 2 | 3 | 4 | 5 | Avg. | Std. |
|----------|-------|-----|-----|-----|-----|-----|------|------|
| A | gpt-4-0613 | 0.732 | 0.707 | 0.732 | 0.713 | 0.738 | 0.724 | 0.013 |
| A | gpt-3.5-turbo-0613 | 0.360 | 0.366 | 0.360 | 0.348 | 0.354 | 0.357 | 0.007 |
| B | gpt-4-0613 | 0.787 | 0.811 | 0.817 | 0.829 | 0.817 | 0.812 | 0.016 |
| B | gpt-3.5-turbo-0613 | 0.348 | 0.354 | 0.348 | 0.335 | 0.348 | 0.346 | 0.007 |
| C | gpt-4-0613 | 0.805 | 0.805 | 0.817 | 0.793 | 0.780 | 0.800 | 0.014 |
| C | gpt-3.5-turbo-0613 | 0.585 | 0.567 | 0.573 | 0.579 | 0.579 | 0.577 | 0.007 |

**Qualitative results**  Figure 11 and Figure 12 illustrate the outcomes of the Architect agent's efforts to design a complex recommender system. These figures showcase the comprehensive system interface design and program call flow. The latter is essential for creating a sophisticated automated system. It is crucial to emphasize the importance of this division of labor in developing an automated software framework.

# D  LIMITATION AND ETHICS CONCERNS

## D.1  LIMITATION

**System side**  At present, our system cannot fully cater to specific scenarios, such as UI and front-end, as we have yet to incorporate such agents and multimodal tools. Furthermore, despite generating the most amount of code among comparable frameworks, it remains challenging to fulfill real-world applications' diverse and complex requirements.

**Human user side**  A key challenge for users is to interrupt the running process of each agent, or set the starting running point (checkpoint) for each agent.

## D.2  ETHICS CONCERNS

**Unemployment and Skill Obsolescence**  MetaGPT enables more people to program in natural languages, thereby making it easier for engineers to get started. Over the years, programming languages have evolved from punched cards to assembly, C, Java, Python, and now natural language. As a result, humans have become more proficient at programming, increasing the demand for programming-related positions. Furthermore, programming with natural language may offer a significantly easier learning curve, making programming more accessible to a broader audience.

**Transparency and Accountability**  MetaGPT is an open-source framework that facilitates interactive communication between multiple agents through natural language. Humans can initiate, observe, and stop running with the highest level of control. It provides real-time interpretation and operation of the natural language, displayed on the screen and logs, ensuring transparency. MetaGPT enhances "natural language programming" capabilities, and human engineers are the users and responsible for the outcomes.

**Privacy and Data Security**  MetaGPT operates locally, ensuring user data privacy and security. It does not collect user data. For interactions with third-party LLMs, such as those by OpenAI, users are encouraged to refer to the respective privacy policies (e.g., OpenAI Privacy Policy). However, we provide the option of open-source LLMs as backends.

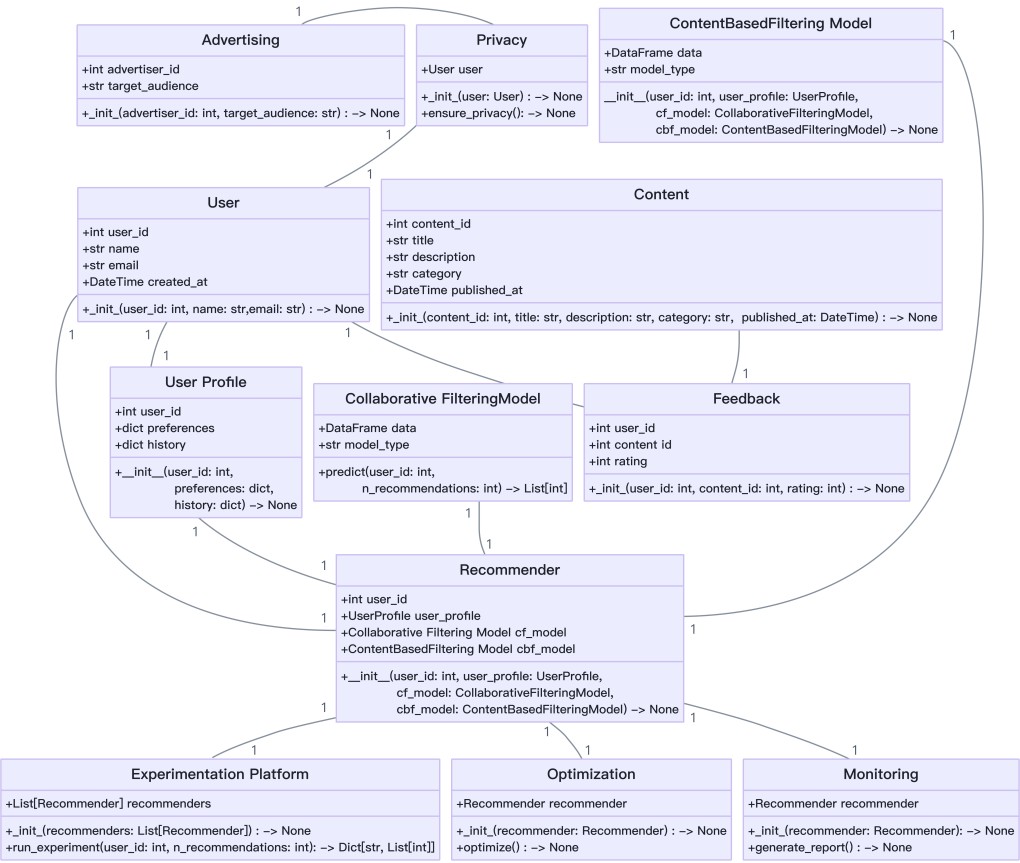

Figure 11: The system interface design for "recommendation engine development" is generated by the *architect* agent (**zoom in for a better view**).

# E   MORE DISCUSSIONS

## E.1   DEEP-SEATED CHALLENGES

MetaGPT also alleviates or solves these challenges with its unique designs:

**Use Context Efficiently**    Two sub-challenges are present. First, unfolding short natural language descriptions accurately to eliminate ambiguity. Second, maintaining information validity in lengthy contexts, enables LLMs to concentrate on relevant data without distraction.

**Reduce Hallucinations**    Using LLMs to generate entire software programs faces code hallucination problems—-including incomplete implementation of functions, missing dependencies, and potential undiscovered bugs, which may be more serious. LLMs often struggle with software generation due to vague task definitions. Focusing on granular tasks like requirement analysis and package selection offers guided thinking, which LLMs lack in broad task solving.

## E.2   INFORMATION OVERLOAD

In MetaGPT, we use a global message pool and a subscription mechanism to address "information overload," which refers to the problem of receiving excessive or irrelevant information. This issue is dependent on specific applications. MetaGPT employs a message pool to streamline communication, ensuring efficiency. Additionally, a subscription mechanism filters out irrelevant contexts, enhancing the relevance and utility of the information. This design is particularly crucial in soft-

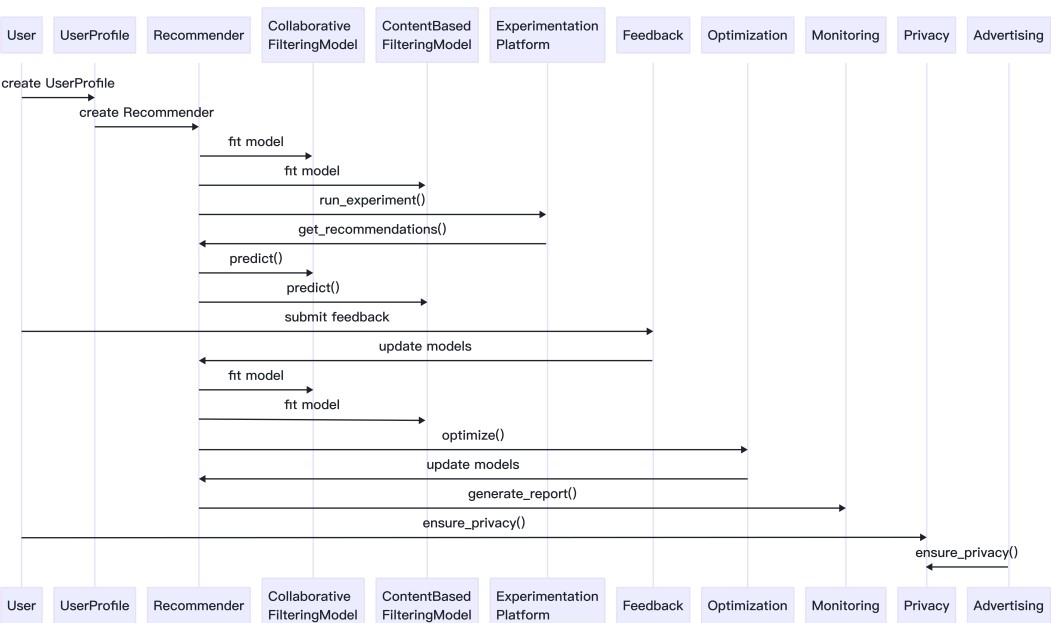

Figure 12: The program call flow for "recommendation engine development" generated by the *architect* agent (**zoom in for a better view**).

ware design scenarios and standard operating procedures (SOPs) where effective communication is essential.

Table 8: Examples of SoftwareDev dataset.

| Task ID | Task | Prompt |
|---|---|---|
| 0 | Snake game | Create a snake game. |
| 1 | Brick breaker game | Create a brick breaker game. |
| 2 | 2048 game | Create a 2048 game for the web. |
| 3 | Flappy bird game | Write p5.js code for Flappy Bird where you control a yellow bird continuously flying between a series of green pipes. The bird flaps every time you left click the mouse. If it falls to the ground or hits a pipe, you lose. This game goes on indefinitely until you lose; you get points the further you go. |
| 4 | Tank battle game | Create a tank battle game. |
| 5 | Excel data process | Write an excel data processing program based on streamlit and pandas. The screen first shows an excel file upload button. After the excel file is uploaded, use pandas to display its data content. The program is required to be concise, easy to maintain, and not over-designed. It uses streamlit to process web screen displays, and pandas is sufficient to process excel reading and display. Please make sure others can execute directly without introducing additional packages. |
| 6 | CRUD manage | Write a management program based on the crud addition, deletion, modification and query processing of the customer business entity. The customer needs to save this information: name, birthday, age, sex, and phone. The data is stored in client.db, and there is a judgement whether the customer table exists. If it doesn't, it needs to be created first. Querying is done by name; same for deleting. The program is required to be concise, easy to maintain, and not over-designed. The screen is realized through streamlit and sqlite—no need to introduce other additional packages. |
| 7 | Music transcriber | Develop a program to transcribe sheet music into a digital format; providing error-free transcribed symbolized sheet music intelligence from audio through signal processing involving pitch and time slicing then training a neural net to run Onset Detected CWT transforming scalograms to chromagrams decoded with Recursive Neural Network focused networks. |
| 8 | Custom press releases | Create custom press releases; develop a Python script that extracts relevant information about company news from external sources, such as social media; extract update interval database for recent changes. The program should create press releases with customizable options and export writings to PDFs, NYTimes API JSONs, media format styled with interlink internal fixed character-length metadata. |
| 9 | Gomoku game | Implement a Gomoku game using Python, incorporating an AI opponent with varying difficulty levels. |
| 10 | Weather dashboard | Create a Python program to develop an interactive weather dashboard. |

Table 9: **Additional results of pure MetaGPT w/o feedback on SoftwareDev.** Averages (Avg.) of 70 tasks are calculated and 10 randomly selected tasks are included. '#' denotes 'The number of', while 'ID' is 'Task ID'.

| ID | Code statistics | | | Doc statistics | | | Cost statistics | | | | Cost of revision | Code executability |
|---|---|---|---|---|---|---|---|---|---|---|---|---|
| | #code files | #lines of code | #lines per code file | #doc files | #lines of doc | #lines per doc file | #prompt tokens | #completion tokens | time costs | money costs | | |
| 0 | 5.00 | 196.00 | 39.20 | 3.00 | 210.00 | 70.00 | 24087.00 | 6157.00 | 582.04 | $ 1.09 | 1. TypeError | 4 |
| 1 | 6.00 | 191.00 | 31.83 | 3.00 | 230.00 | 76.67 | 32517.00 | 6238.00 | 566.30 | $ 1.35 | 1. TypeError | 4 |
| 2 | 3.00 | 198.00 | 66.00 | 3.00 | 235.00 | 78.33 | 21934.00 | 6316.00 | 553.11 | $ 1.04 | 1. lack @app.route('/') | 3 |
| 3 | 5.00 | 164 | 32.80 | 3.00 | 202.00 | 67.33 | 22951.00 | 5312.00 | 481.34 | $ 1.01 | 1. PNG file missing 2. Compile bug fixes | 2 |
| 4 | 6.00 | 203.00 | 33.83 | 3.00 | 210.00 | 70.00 | 30087.00 | 6567.00 | 599.58 | $ 1.30 | 1. PNG file missing 2. Compile bug fixes 3. pygame.surface not initialize | 3 |
| 5 | 6.00 | 219.00 | 36.50 | 3.00 | 294.00 | 96.00 | 35590.00 | 7336.00 | 585.10 | $ 1.51 | 1. dependency error 2. ModuleNotFoundError | 4 |
| 6 | 4.00 | 73.00 | 18.25 | 3.00 | 261.00 | 87.00 | 25673.00 | 5832.00 | 398.83 | $ 0.90 | 0 | 4 |
| 7 | 4.00 | 316.00 | 79.00 | 3.00 | 332.00 | 110.67 | 29139.00 | 7104.00 | 435.83 | $ 0.92 | 0 | 4 |
| 8 | 5.00 | 215.00 | 43.00 | 3.00 | 301.00 | 100.33 | 29372.00 | 6499.00 | 621.73 | $ 1.27 | 1. tensorflow version error 2. model training method not implement | 2 |
| 9 | 5.00 | 215.00 | 43.00 | 3.00 | 270.00 | 90.00 | 24799.00 | 5734.00 | 550.88 | $ 1.27 | 1. dependency error 2. URL 403 error | 3 |
| 10 | 3.00 | 93.00 | 31.00 | 3.00 | 254.00 | 84.67 | 24109.00 | 5363.00 | 438.50 | $ 0.92 | 1. dependency error 2. missing main func. | 4 |
| Avg. | 4.71 | 191.57 | 42.98 | 3.00 | 240.00 | 80.00 | 26626.86 | 6218.00 | 516.71 | $1.12 | 0.51 (only consider item scored 2, 3 or 4) | 3.36 |

