# OpenReview forum: "MetaGPT: Meta Programming for A Multi-Agent Collaborative Framework"
_ICLR.cc/2024/Conference — ICLR 2024 oral_

### Official Review · Reviewer_uj5i · 2023-10-31

**Soundness:** 4 excellent
**Presentation:** 4 excellent
**Contribution:** 3 good
**Rating:** 8
**Confidence:** 4

**Summary:**

This paper introduces MetaGPT, an innovative meta-programming framework for multi-agent collaborations based on LLM, which encodes Standardized Operating Procedures (SOPs) into prompt sequences for more streamlined workflows. It selects a group of agents as a simulated software company, to generate a variety of code-based softwares. Through extensive experiments, MetaGPT achieves state-of-art performance on multiple code benchmarks HumanEva, MBPP and a software development benchmark SoftwareDev.

**Strengths:**

1. The idea of encodes SOPs of software development into LLM-based multi-agent systems is very interesting and also pracitical to use.
2. The framework is very sound and solid, with Specialization of Roles, PRD workflow across Agents, Structured Communication for complex tasks, and a compute-efficient Message Pool mechanism with both global memory and Subscription Mechanism. It also introduces an executive feedback mechanism to enhance code generation quality during runtime.
3. MetaGPT achieves state-of-art performance on multiple benchmarks such as HumanEva, MBPP and a software development benchmark SoftwareDev. It opens new possibility for the software development paradigm.

**Weaknesses:**

1. Most of the experiment are on GPT4, which is expensive to access, how is the performance on the benchmarks or real development demands when used with open-source LLMs? Can you share some insight on which abilities of the LLMs matters most for the success of using this multi-agent framework and how to choose proper LLMs for use?

**Questions:**

1. Does the framework have strong generalizability when switched to GPT 3.5 or other open-source models? how far can the framework or practical use go with open-source models?
2. In table 3, In this multi-agent collaboration with different roles, do you have some qualitive analysis or case study to show where each role contributes to the final performance, except for an executability score? Can you provide more detailed analysis?
3. Whether the framework can also solve problems other than code, such as math and QA in Autogen? Is there other challenges for these kinds of problems for the framework?

---

> ### Author Response · Authors · 2023-11-19
> **Thanks for your review! Authors' feedback.**
>
> We appreciate your valuable time and positive comments (i.e., **the idea of encoding SOPs of software development into LLM-based multi-agent systems is very interesting and also practical to use**, **the framework is very sound and solid**, **it opens new possibilities for the software development paradigm.**). In the following, we will carefully respond to your questions.
>
> ---
>
> **Q1.  Can you share some insight on which abilities of the LLMs matter most for the success of using this multi-agent framework and how to choose proper LLMs for use?**
>
> In our experience, coding tasks need GPT-4, while GPT-3.5 suffices for other tasks without obvious performance impact. We recommend using models fine-tuned for specific skills in future developments. For example, a model specialized in code generation can be an engineer's LLM, optimizing cost and role performance.
>
> ---
>
> **Q2. Most of the experiments are on GPT4, which is expensive to access, how is the performance on the benchmarks or real development demands when used with open-source LLMs?  Does the framework have strong generalizability when switched to GPT 3.5 or other open-source models? how far can the framework or practical use go with open-source models?**
>
> Following your suggestions, we evaluated the HumanEval & MBPP datasets and a few SoftwareDev tasks to assess using GPT-3.5 and Deepseek Coder 33B as backends. The results indicate that although MetaGPT can complete tasks with these LLMs, using GPT-4 as the backend yields superior performance.
>
>
> |Model|HumanEvel|MBPP|
> |:--------------------|:----------------|:----------------|
> |GPT-3.5|$48.1$|$52.2$|
> |MetaGPT (w/ GPT-3.5)| $62.8 (+14.7)$|$74.7 (+21.5)$|
> |GPT-4| $67.0$|-|
> |MetaGPT (w/ GPT-4)|$85.9 (+18.4)$|$87.7$|
>
>
> | Model| Open source|Running time(s)|Total Code Lines|Executability |Human Revision Cost(times)|
> |---|---|:---:|:---:|:---:|:---:|
> |MetaGPT (w/ GPT-3.5)|✗|$75.18$|$161.6$|$2.8$|$2.4$|
> |MetaGPT (w/ GPT-4)|✗|$552.94$|$178.2$|$3.8$|$1.2$|
> |MetaGPT (w/ Deepseek Coder 33B)|✓|$1186.20$|$120.2$|$1.4$|$2.6$|
>
> ---
>
>
> **Q3: In Table 3, In this multi-agent collaboration with different roles, do you have some qualitative analysis or case study to show where each role contributes to the final performance, except for an executability score? Can you provide more detailed analysis?**
>
> Thank you for your comment. To illustrate, we'll use the task **'write a python3 GUI app which allows drawing images on it'** as an example (the intermediate processes involving full team members are detailed in Appendix B). Here, we provide a qualitative analysis of each role's contribution to the final performance.
>
> Here is the summary of the impact of removing certain agent roles:
>
> - **Without Architect:**
>   - An absence of detailed system design leads to structural and functional issues in coding, which affects the robustness of final product.
>   - Task decomposition lacks sufficient granularity.
> - **Without ProjectManager:**
>   - Missing module dependencies result in incomplete functionality and additional bugs.
> - **Without ProductManager:**
>   - Lack of function design and plan results in incomplete design and missing functionality, directly impacting the total number of files and generated code.
>
>
> ---
>
> **Q4. Whether the framework can also solve problems other than code, such as math and QA in Autogen? Is there other challenges for these kinds of problems for the framework?**
>
> MetaGPT can address a variety of domain issues with minor implementation changes. While our current implementation (of submission) may not be ideal for specific tasks like Math and QA, we can quickly create specialized agents for these areas. Benefiting from MetaGPT's cost-effectiveness and adaptability in task modeling, we've successfully used it to replicate applications, including games like Minecraft [1] and Werewolf [2], and in roles like research assistants.
>
> [1] Wang G, Xie Y, Jiang Y, et al. "Voyager: An open-ended embodied agent with large language models". arXiv:2305.16291, 2023.
>
> [2] Xu Y, Wang S, Li P, et al. "Exploring large language models for communication games: An empirical study on werewolf". arXiv:2309.04658, 2023.
>
> ---
>
> **We appreciate your time in reading our feedback and look forward to further discussion and your suggestions.**

---

### Official Review · Reviewer_itRX · 2023-10-31

**Soundness:** 4 excellent
**Presentation:** 3 good
**Contribution:** 4 excellent
**Rating:** 8
**Confidence:** 3

**Summary:**

The paper introduces MetaGPT, a meta-programming framework designed to enhance the efficiency and accuracy of large language model (LLM)-based multi-agent systems. By incorporating Standardized Operating Procedures (SOPs) into prompt sequences and adopting an assembly line approach, MetaGPT enables agents to specialize in diverse roles and collaboratively solve complex tasks with reduced errors.

I believe this paper is interesting and makes a contribution by introducing an instrumental tool for automated code generation. I have several questions that I believe need to be addressed before publication. Aside from the questions below, my biggest concern is the relevance of the paper to Learning Representations. This paper mostly seems to be a nice fit for a journal focused on systems and applications.

- It appears that this system needs existing expertise and strategy, correct? How primal/high-level the initial instructions can be? For instance, is a novice user going to have a hard time using this because they don’t have enough knowledge about breaking an initial task into sub-steps?

- Does the level of initial input from human affect the performance? For instance, if the human gives a very high-level input vs a more detailed, structured input instruction?

- The human user side of the system is never discussed, although I believe it’s very critical to understand the requirements, skills, knowledge, etc. the system imposes on or requires from a human user. The initial task or input instructions from the user are also never discussed.

- The paper uses the term collaborative multi-agent systems, but later mentions that they define multi-agent frameworks as LLM-based multi-agent systems. I think the term collaborative multi-agent systems is very broad and comprehensive and it could be misleading to readers. The ‘LLM-based’ attribute of the multi-agent systems must be reflected in Title. Also, this leads to some misleading sentences such as “Most of current multi-agent frameworks utilize natural language as a communication interface.” Without using the term LLM-based multi-agent frameworks, this sentence is incorrect.

- Is the shared communication/message pool interpretable to humans? Is it accessible by humans during or after the process?

- Would the system ever need human interaction, or further input from human to address its potential questions?

- There are multiple agents introduced in the system each with different roles, however, it is never discussed how are these specialized agents built or trained, algorithmically. Are they using specifically trained models?

- Can there be multiple agents with the same role? i.e., multiple engineers that would work on different parts of implementations in parallel? How’s the parallelization process performed? For example, if multiple engineers will be working different parts of the problem designated by the architect, how will they choose which part to work on? Is there priority assigned, or is there a decision-making problem being solved? Sometimes the order by which a problem is solved could significantly affect its efficiency.

- What are some of the limitations of the system at its current stage? Limitations, both in the system and on the human user side must be discussed.

At current states I vote weak reject (mostly due to relevance), although the system seems to be sound and working. I need to see more discussions and revisions as suggested above, as well as suggested by my fellow reviewers to make this an ICLR-ready paper. I’d be happy to increase my score when authors satisfactorily addressed the questions and comments.

**Strengths:**

See above.

**Weaknesses:**

See above.

**Questions:**

See above.

---

> ### Author Response · Authors · 2023-11-19
> **Thanks for your review! Authors' feedback [1/2].**
>
> We appreciate your time and positive feedback (for examples, **"I believe this paper is interesting and makes a contribution by introducing an instrumental tool for automated code generation. I have several questions that I believe need to be addressed before publication.")**. We will carefully answer your questions one by one.
>
> ---
>
> **Q1. Aside from the questions below, my biggest concern is the relevance of the paper to Learning Representations.**
>
> Our work aligns with the objectives of the tolerant ICLR community and focuses on the topic [1] – applications in robotics, autonomy, and planning. Similar papers have been accepted in related communities [2,3,4].
>
> [1] https://iclr.cc/Conferences/2024/CallForPapers
>
> [2] Zeng A, Attarian M, Ichter B, et al. "Socratic models: Composing zero-shot multimodal reasoning with language". In ICLR, 2023.
>
> [3] Li G, Hammoud H A A K, Itani H, et al. "Camel: Communicative agents for" mind" exploration of large scale language model society". In NeurIPS, 2023.
>
> [4] Yao S, Zhao J, Yu D, et al. "React: Synergizing reasoning and acting in language models". In ICLR, 2022.
>
> ---
>
> **Q2**. **(1) It appears that this system needs existing expertise and strategy, correct? (2) How primal/high-level the initial instructions can be? For instance, is a novice user going to have a hard time using this because they don’t have enough knowledge about breaking an initial task into sub-steps?**
>
> (1) For developers, it's correct that multi-agent frameworks typically require humans to pre-define the agents (which need existing expertise and strategy). However, for users, we have already completed these definitions, enabling users to generate software directly without significant barriers.
>
> (2) There is no need to break down an initial task; users can use very basic instructions, such as 'design a Flappy Bird game' or 'write a Python3 GUI app where you can draw an image', etc. The subsequent processes are automated by MetaGPT, making it user-friendly. Additional examples can be found in rows 0-2, 4, 9-10 of Table 5 (Appendix), with their corresponding performances outlined in Table 6 (Appendix).
>
> ---
>
> **Q3. Does the level of initial input from human affect the performance? For instance, if the human gives a very high-level input vs a more detailed, structured input instruction?**
>
> Thanks for these valuable comments. We conduct additional experiments to verify your questions. Here, we first give two examples of prompts:
> * **High-level prompt**: `Create a brick breaker game.`
> * **Detailed prompt**: `Creating a brick breaker game. In a brick breaker game, the player typically controls a paddle at the bottom of the screen to bounce a ball towards a wall of bricks. The goal is to break all the bricks by hitting them with the ball.`
>
> We select 5 tasks from SoftwareDev, and construct detailed prompts for them. Here are the experimental results:
>
> |$\textbf{Model}$|$\textbf{Num. of word}$|$\textbf{Running time(s)}$|$\textbf{Token usage}$|$\textbf{Total Code Lines}$|$\textbf{Executability}$|$\textbf{Productivity}$|$\textbf{Human Revision Cost}$|
> |---|---|---|---|---|---|---|---|
> |$\textbf{High-level prompt}$|$13.2$|$552.9$|$28384.2$|$178.2$|$3.8$|$163.8$|$1.2$|
> |$\textbf{Detailed prompt}$  |$42.2$|$567.8$|$29657.0$|$257.0$|$4.0$|$118.0$|$1.6$|
>
> We observe that: **detailed prompts lead to better software projects with lower productivity ratios because of clearer requirements and functions, while simple inputs can still generate good enough software using MetaGPT with an executability rating of 3.8, which is comparable to the detailed prompt scenario.** (Note that, Productivity = Token usage / Total Code Lines. The lower this ratio, the better.)
>
> ---
>
> **Q4. The human user side of the system is never discussed, although I believe it’s very critical to understand the requirements, skills, knowledge, etc. the system imposes on or requires from a human user. The initial task or input instructions from the user are also never discussed.**
>
> Thanks, this is a very valuable comment. We will add our responses in **Q3** and **Q4** to fill this gap in the manuscript.
>
> ---
>
> **Q5. The ‘LLM-based’ attribute of the multi-agent systems must be reflected in the Title. Also, this leads to some misleading sentences such as “Most of current multi-agent frameworks utilize natural language as a communication interface.**
>
> Thank you for pointing out this issue. As pointed out in related work: ``We refer to multi-agent frameworks as LLM-based multi-agents and distinguish them from previous settings (Schmidhuber, 1999; Busoniu et al., 2008)''. **We agree with you that we need to change the illustration of the misleading sentence to "Most of the current LLM-based multi-agent frameworks..."** For the title, we will use "MetaGPT: Meta Programming for A Multi-Agent Collaborative Framework" to make it clear that MetaGPT is one (and one kind) of the multi-agent collaborative frameworks.

---

> ### Author Response · Authors · 2023-11-19
> **Thanks for your review! Authors' feedback [2/2].**
>
> **Q6. Is the shared communication/message pool interpretable to humans? Is it accessible by humans during or after the process?**
>
> During the running, **humans can actively watch the inputs and outputs of each role in the terminal.** MetaGPT also automatically saves the intermediate files, such as PDFs, PNGs, Python code, etc., during the execution. Humans can easily access them after the process.
>
> ---
>
> **Q7. Would the system ever need human interaction, or further input from human to address its potential questions?**
>
> This is a good question. In our design, **in order to strengthen full automation (honestly speaking, it is more appealing)**, we proactively ignored human interactions. However, **we also provide an interface, and humans can easily interact with MetaGPT if they want**, such as helping to debug the code or revise the PRD. In terms of techniques, it is not hard to achieve.
>
> ---
>
> **Q8. There are multiple agents introduced in the system each with different roles, however, it is never discussed how are these specialized agents built or trained, algorithmically. Are they using specifically trained models?**
>
> Regarding the construction of agents, we define each role's profile, name, goal, and constraints, while also initializing their specific context and skills, which is stated in the first paragraph Sec. 3.1 and illustrated in Figure 2 (right). We employ GPT-4 as the only trained model and the default LLM backend for all agents. It is noteworthy that, in terms of experiment cost, other open-source LLMs can be used flexibly in MetaGPT, such as using CodeX [5] for Engineers.
>
> [5] Chen M, Tworek J, Jun H, et al. "Evaluating large language models trained on code". arXiv:2107.03374, 2021.
>
> ---
>
> **Q9. Can there be multiple agents with the same role? i.e., multiple engineers that would work on different parts of implementations in parallel? How’s the parallelization process performed? For example, if multiple engineers will be working different parts of the problem designated by the architect, how will they choose which part to work on? Is there priority assigned, or is there a decision-making problem being solved? Sometimes the order by which a problem is solved could significantly affect its efficiency.**
>
> Thank you for bringing up a professional question. In our earlier version, we incorporated a simplified approach to handle multiple engineers working on the same project. Although different agents could write different code files simultaneously, it didn't necessarily guarantee an improvement in the overall output quality of the project. We agree that processing dependencies between files and modules poses a more complex decision-making problem.
>
> ---
>
> **Q10. What are some of the limitations of the system at its current stage? Limitations, both in the system and on the human user side must be discussed.**
>
> 1) **System side**: At present, our system cannot fully cater to specific scenarios, such as UI and front-end, as we have yet to incorporate such agents and multimodal tools. Furthermore, despite generating the most amount of code among comparable frameworks, it remains challenging to fulfill real-world applications' diverse and complex requirements.
>
> 2) **Human user side**: A key challenge for users is to interrupt the running process of each agent, or set the starting running point (checkpoint) for each agent.
>
> ---
>
> **We appreciate your time in reading our feedback and look forward to further discussion and your suggestions.**

---

> > ### Comment · Reviewer_itRX · 2023-11-20
> > **Response to Authors**
> >
> > Thank you to authors for clarifying. I also appreciate the additional experiments and results which add to the value of their work. I also read the comments by my fellow reviewers and overall, I'm positively satisfied and vote for acceptance. I raised my score accordingly. Congratulations to the authors for their great work and, good luck!

---

> > > ### Author Response · Authors · 2023-11-20
> > > **Thank you!**
> > >
> > > Thank you for your practical tips and professional advice, which helped us make our paper better, more impactful, and more thoroughly researched, ultimately strengthening our overall work. We also appreciate your fast reply and improve the score!

---

### Official Review · Reviewer_WkUQ · 2023-11-11

**Soundness:** 2 fair
**Presentation:** 3 good
**Contribution:** 2 fair
**Rating:** 3
**Confidence:** 4

**Summary:**

MetaGPT is a framework designed for programming in multi-agent collaborative environments, specifically utilizing the Large Language Model (LLM). Its main contribution is the integration of human workflows into LLM-based multi-agent collaboration, which is effective for complex software engineering tasks. The framework assigns different roles to the GPT model, similar to a pipeline in software development, including roles such as product manager, architect, and engineer. Its main advantage lies in generating coherent solutions and decomposing complex tasks through the collaboration of multiple AI agents.

**Strengths:**

1. Problem-solving: MetaGPT can solve complex tasks by using different AI agents to handle different parts of the problem.

2. Clear solutions: MetaGPT provides sensible solutions that favor tasks that require different elements to work together seamlessly.

3. Flexibility: MetaGPT can be used for different tasks, and provide multiple outputs ranging from project plans to technical documentation.

**Weaknesses:**

1. Methodological innovation: A key issue with MetaGPT is the lack of innovation in its methodology. While it effectively utilizes Large Language Models (LLMs) in multi-agent systems, this approach may not be significantly different from existing approaches. Or MetaGPT is different from other methods in a trivial way, I don't really see their differences as being significant and requiring exploration with greater depth.

2. Fairness of experimental comparisons: The comparison methods used to assess the effectiveness of MetaGPT do not appear to be fully equivalent to what MetaGPT offers. Such differences may lead to biased or misleading conclusions about their superiority or efficiency.


3. Experimental validation of statements: Current experiments may not adequately validate the authors' claims about the efficiency and robustness of MetaGPT.

**Questions:**

1. My primary concern is that MetaGPT lacks methodological and theoretical innovation. To gain a more innovative edge, MetaGPT could deeply explore what deep-seated challenges in integrating newer AI techniques or unique collaboration strategies make it more clearly distinguishable from other LLM-based frameworks. (Honestly, I think MetaGPT is more suited for system demonstration conferences or tracks)

2. In the Introduction, "MetaGPT stands out as a special solution that allows efficient meta-programming through a well-organized and specialized group of agents". How does the author define these professional concepts? For example, "efficient" and "well-organized".1) I haven't seen any evidence or convincing analysis of why it can be "efficient", nor have I seen experimental validation.2) How do you define "specialized group"? This seems trivial and can be varied.

3. In the Introduction, "MetaGPT achieves a 100% task completion rate, further demonstrating the robustness and efficiency of our design", and "Our innovative integration of human-like SOPs throughout MetaGPT’s design significantly enhances its robustness". 1) What is the definition of "robustness" here? I didn't see any discussion of "robustness" except in the Introduction section. 2) What are the deeper methodological challenges of integrating human-like SOPs?

4. In the Related Work, "In this paper, we identify the main challenges in multi-agent cooperation: maintaining consistency, avoiding unproductive cycles, and controlling beneficial interactions". There seems to be inconsistency in the description of the research challenges between the "Related Work", the "Introduction" and the rest of the paper. Furthermore, is it trivial to maintain consistency in MetaGPT?" What are the professional definitions of "unproductive cycle" and "beneficial"? Are there experimental results to validate these advantages?

5. In the Experiments, "We compare recent domain-specific LLMs in the code generation field ..." Is it fair to compare with these LLMs?

6. In the Experiments, "We modified certain role-based prompts in order to instruct the MetaGPT framework to generate individual functions instead of entire classes" 1) Is it fair to compare with these frameworks? LangChain and AutoGPT, for example, which are not specific to the tasks involved in this commit. 2) How can we ensure that " prompt modification" does not lead to unfair comparisons?

7. Figure 4. What are the results of multiple attempts and are they stable? Were there any problems, like robustness?

8. In Section 4.2, what is "the cost of human revision" and how can it be rigorously measured?

**Details Of Ethics Concerns:**

This submission does not discuss any potential ethical issues, which I think is uncritical, especially for generative AI technologies. There are potential privacy, security, and bias issues that accompany the collection, processing, and use of data, exchange between agents, communication, and so on, in open-ended tasks with multi-person collaboration. For example, I perceive at least the following concerns and considerations.

1. Unemployment and skill obsolescence: Automation of complex tasks in software engineering and other fields may lead to job losses. Professionals in these fields may need to adapt and acquire new skills to remain relevant. How to manage and communicate these changes is an ethical issue.

2. Transparency and accountability: As AI takes on more complex collaborative tasks, it becomes critical to maintain transparency in decision-making. It should be clear how MetaGPT arrives at solutions and who is accountable for those decisions, especially in critical applications.

3. Privacy and data security: The use of large-scale language models and AI in processing potentially sensitive information raises concerns about data privacy and security. Ensuring that user data is protected and used ethically is a key consideration.

---

> ### Author Response · Authors · 2023-11-19
> **Thanks for your review! Authors' feedback [1/3].**
>
> We appreciate your valuable time, insights, and highlight our strengths (i.e., **problem-solving**, **clear solutions**, and **flexibility**). We'll address your concerns in our following response.
>
> ---
>
> **Q1 (original W1). Methodological innovation: How does MetaGPT differ from other methods?**
>
> > 1) **Agents in SOPs**: MetaGPT pioneers the introduction of SOPs (Standard Operating Procedures), a departure from works like CAMEL [2], which don't discuss task-specific labor division. Unlike generative agents [1] that simulate open-world human behavior freely, MetaGPT focuses more on applying valuable human practice and better organizing multiple agents.
> > 2) **Structured communication v.s. Chat-based communication**: Sec 3.2 highlights how MetaGPT, unlike other chat-based communication models such as ChatDev [3], Self-collaboration [4], and CAMEL [2], introduces structured communication. MetaGPT limits the range of message content, focusing agents on relevant information and avoiding meaningless conversational chat. Consequently, this approach reduces token usage and enhances task success rates.
>
> [1] Park, Joon Sung, et al. "Generative agents: Interactive simulacra of human behavior." In UIST, 2023.
>
> [2] Li, Guohao, et al. "CAMEL: Communicative Agents for" Mind" Exploration of Large Language Model Society." In NeurIPS, 2023.
>
> [3] Qian, Chen, et al. "Communicative agents for software development." arXiv:2307.07924, 2023.
>
> [4] Dong Y, Jiang X, Jin Z, et al. "Self-collaboration Code Generation via ChatGPT." arXiv:2304.07590, 2023.
>
> ---
>
> **Q2 (Original Q1). Deep-seated challenges.**
>
> MetaGPT also alleviates or solves these challenges with its unique designs:
> 1) **Use Context Efficiently**: Two sub-challenges are present. First, unfolding short natural language descriptions accurately to eliminate ambiguity. Second, maintaining information validity in lengthy contexts, enables LLMs to concentrate on relevant data without distraction.
> 2) **Reduce Hallucinations**:  Using LLMs to generate entire software programs faces code hallucination problems—-including incomplete implementation of functions, missing dependencies, and potential undiscovered bugs, which may be more serious than hallucinations in natural language generation. Deep-seated challenges in task decomposition and translating from natural language to code language persist. LLMs often struggle with software generation due to vague task definitions. Focusing on granular tasks like requirement analysis and package selection offers guided thinking, which LLMs lack in broad task solving.
>
> ---
>
> **Q3 (Original W2, Q5, Q6). The comparison methods used to assess the effectiveness of MetaGPT do not appear to be fully equivalent to what MetaGPT offers. **(1)** "We compare recent domain-specific LLMs in the code generation field ..."; **(2)** "We modified certain role-based prompts in order to instruct the MetaGPT framework to generate individual functions instead of entire classes. Is it fair to compare with these frameworks? LangChain and AutoGPT? How can we ensure that prompt modification does not lead to unfair comparisons?"**
>
> Thanks for pointing these out, but these are misunderstandings.
> **(1) Both the baselines and MetaGPT are conducted under the same settings.** We acknowledge that comparing MetaGPT with single LLMs may seem partially unfair. However, our goal is to help readers quickly grasp the advancements in this field, where both LLMs (PaLM Code vs. GPT-4) and methodologies (Codex vs. Codex+CodeT) are essential. The most important point here is "When MetaGPT collaborates with GPT-4, it substantially improves the Pass@1 in the HumanEval benchmark, exceeding the performance of GPT-4 alone" as stated in the manuscript.
> **(2) In the SoftwareDev dataset, we maintain fully equivalent settings for a fair comparison with AutoGPT, LangChain, and ChatDev.** In HumanEval and MBPP, we've slightly modified the prompts to align with response format requirements. These modifications aim to address format-specific issues (i.e., Python problems). To enhance clarity in our manuscript, we propose adding 'In HumanEval and MBPP' at the beginning of Sec 4.1.
>
> ---
>
> **Q4 (Original Q3). Statements: "MetaGPT achieves a 100% task completion rate, further demonstrating the robustness and efficiency of our design", and "Our innovative integration of human-like SOPs throughout MetaGPT's design significantly enhances its robustness". What is the definition of "robustness" here?**
>
> **Robustness** refers to the ability to generate software codes that can complete various tasks with consistent quality and adhere to specific designs. （Figure 5 & Table 6）Existing experiments support these claims. For example: 1) In Figure 5, we display many demos generated by MetaGPT, demonstrating its robustness to generate different software. 2) In Table 4 & Table 6 (Appendix), we have shown that MetaGPT can generate high-quality software aligned with expectations.
>
> ---

---

> ### Author Response · Authors · 2023-11-19
> **Thanks for your review! Authors' feedback [2/3].**
>
> **Q5 (Original Q2). Statements: "MetaGPT stands out as a special solution that allows efficient meta-programming through a well-organized and specialized group of agents". How does the author define these professional concepts? For example, "efficient" and "well-organized". **(1)** I haven't seen any evidence or convincing analysis of why it can be "efficient", nor have I seen experimental validation. **(2)** How do you define "specialized group"?**
>
> **(1) Existing experiments validate our claims.** In Table 1, we show "efficiency" with lower Running Times (762 vs. 541 seconds) than ChatDev, using identical instructions for software generation. We also report higher Productivity (token usage/total code lines), with our figures at 248.9 vs. 124.3, using half as many tokens per code line.
>
> |Model|Running Times (/seconds)|Productivity (token usage/total code lines)|
> |---|---|---|
> |ChatDev|$762$|$248.9$|
> |MetaGPT|$541$|$124.3$|
>
> **(2)** In our context, a **'specialized group'** refers to a group of agents where each agent is assigned specific roles with unique tools and skills. This ensures efficient task execution and collaboration. **For instance, the ProductManager can summarize instructions and write product requirement documents (as in Figure 1), whereas the Engineer focuses on code execution and debugging with coding tools.** In MetaGPT, different roles can also output specialized files in PDF, PNG, and Python formats, and users can directly view them.
>
> ---
>
> **Q6 (Original Q4). Is it trivial to maintain consistency in MetaGPT? What are the professional definitions of "unproductive cycle" and "beneficial"? Are there experimental results to validate these advantages?**
>
> * 1) **Consistency:** Maintaining consistency is not trivial and presents a challenge in many applications, as highlighted in work [1,2,3]. MetaGPT addresses this by breaking down tasks into granular sub-tasks, enabling each agent to gain a deeper understanding of their specific responsibilities. This approach is crucial, particularly in function design and coding requirements for downstream agents (e.g., Engineer), to prevent misunderstandings or error propagation during collaboration using LLMs.
> * 2) **Unproductive cycles:** In work [2], it's noted that two cooperating agents often face challenges like the 'Assistant Repeats Instruction' problem or the 'Infinite Loop of Messages'. These issues emerge from repetitive, context-less conversations leading to unproductive exchanges, such as repeatedly saying "Goodbye". Reference [4] discusses an issue where agents become stuck in a loop while addressing complex or ambiguous tasks, chaining thoughts in a behavior-like pattern. We can improve the quality of code generation by reducing unproductive cycles through SOPs, as described in Table 1.
> * 3) **Beneficial interactions:** Previous work [5] highlights the requirements engineering in software development, particularly abstraction and decomposition, significantly improving code generation. Constraints also improve LLMs' understanding of fuzzy, abstract, and multitasking semantics [6]. Thus, we introduced a structural communication protocol to assist agents in task decomposition and semantic understanding, aiming to minimize ambiguities and errors, fostering beneficial interactions.
>
> When reproduced to other frameworks, we found that these issues occur frequently. The SOPs and communication mechanisms in MetaGPT can alleviate these issues, thus achieving better performances (see Table 1, Table 2, Table 4).
>
> [1] Elazar Y, Kassner N, Ravfogel S, et al. "Measuring and improving consistency in pretrained language models". TACL, 2021.
>
> [2] Li, Guohao, et al. "CAMEL: Communicative Agents for" Mind" Exploration of Large Language Model Society." In NeurIPS, 2023.
>
> [3] Wang X, Wei J, Schuurmans D, et al. "Self-consistency improves chain of thought reasoning in language models". arXiv:2203.11171, 2022.
>
> [4] Talebirad Y, Nadiri A. "Multi-Agent Collaboration: Harnessing the Power of Intelligent LLM Agents". arXiv:2306.03314, 2023.
>
> [5] Jiang X, Dong Y, Wang L, et al. "Self-planning code generation with large language model". arXiv:2303.06689, 2023.
>
> [6] Shin R, Lin C H, Thomson S, et al. "Constrained language models yield few-shot semantic parsers". arXiv:2104.08768, 2021.
>
>
> ---
>
> **Q7 (Original Q8). In Section 4.2, what is "the cost of human revision" and how can it be rigorously measured?**
>
> **Human revision** refers to times of manual code corrections, which tackle problems like package import errors, incorrect class names, or incomplete reference paths. Typically, each correction involves up to 3 lines of code. To minimize subjectivity, we enlisted three individuals (paid at a rate equivalent to a minimum of 15 USD per hour) to document their debugging process.
>
> ---

---

> ### Author Response · Authors · 2023-11-19
> **Thanks for your review! Authors' feedback [3/3].**
>
> **Q8 (Original Q7): Figure 4. What are the results of multiple attempts and are they stable? Were there any problems, like robustness?**
>
> 1) **According to this comment, we conduct GPT-4 and GPT-3.5 experiments on HumanEval and MBPP.** For GPT-4, upon reviewing the results, we observe that MetaGPT has a lower standard deviation of 0.498% for HumanEval, while MBPP experiences a higher standard deviation of 0.862%. Most variation in MBPP results from inaccurate test case generation due to a lack of input examples, resulting in passing tests but failing actual results. MetaGPT can produce robust results for these benchmarks.
>
> 2) **Compared to GPT-4, GPT-3.5-turbo has a higher standard deviation (more unstable)** in both HumanEval and MBPP.
>
> |||GPT-4||||GPT-3.5|||
> |:---:|:---:|:---:|:---:|:---:|:---:|:---:|:---:|:---:|
> ||HumanEval|| MBPP-sanitized ||HumanEval|| MBPP-sanitized||
> ||Attempts|Pass@$1$(%)|Attempts|Pass@$1$(%)|Attempts|Pass@$1$(%)|Attempts|$Pass@$1$(%)|
> ||$1$|$86.0$|$1$|$87.6$|$1$|$61.6$|$1$|$75.9$|
> ||$2$|$84.8$|$2$|$86.7$|$2$|$64.0$|$2$|$74.5$|
> ||$3$|$85.4$|$3$|$85.5$|$3$|$62.8$|$3$|$73.8$|
> ||$\textbf{avg}$(%)|$85.4$|$\textbf{avg}$(%)|$86.6$|$\textbf{avg}$(%)|$62.8$|$\textbf{avg}$(%)|$74.7$|
> ||$\textbf{std}$(%)|$0.490$|$\textbf{std}$(%)|$0.860$|$\textbf{std}$(%)|$0.980$|$\textbf{std}$(%)|$0.873$|
>
>
> ---
>
> **Q9 (Original Details Of Ethics Concerns). Discuss any potential ethical issues.**
>
>
> Thanks for pointing this out. We will include an additional potential ethical section in the Appendix:
> 1) **Unemployment and Skill Obsolescence**: MetaGPT enables more people to program in natural languages, thereby making it easier for engineers to get started. Over the years, programming languages have evolved from punched cards to assembly, C, Java, Python, and now natural language. As a result, humans have become more proficient at programming, increasing the demand for programming-related positions. Furthermore, programming with natural language may offer a significantly easier learning curve, making programming more accessible to a broader audience.
> 2) **Transparency and Accountability**: MetaGPT is an open-source framework that facilitates interactive communication between multiple agents through natural language. Humans can initiate, observe, and stop running with the highest level of control. It provides real-time interpretation and operation of the natural language, displayed on the screen and logs, ensuring transparency. MetaGPT enhances "natural language programming" capabilities, and human engineers are the users and responsible for the outcomes.
> 3) **Privacy and Data Security**: MetaGPT operates locally, ensuring user data privacy and security. It does not collect user data. For interactions with third-party LLMs, such as those by OpenAI, users are encouraged to refer to the respective privacy policies (e.g., OpenAI Privacy Policy). However, we provide the option of open-source LLMs as backends.
>
> ---
>
> **We appreciate your time in reading our feedback and look forward to further discussion and your suggestions.**

---

### Comment · Area_Chair_u7nd · 2023-11-11
**Additional official review**

This note is from the area chair who is also acting as a reviewer for this paper since we didn't end up getting three reviews out of the usual process.


**Soundness:** 4 excellent

**Presentation:** 4 excellent

**Contribution:** 3 good

**Review:**

This paper is about decomposing a complex task into subtasks to be performed by independent language agents to be organized in some way such as, eg. a software company. This is a really nice idea whose time has clearly come. I’ve noticed several versions of the basic idea appear over the last few months (all so recent that it would not be fair to consider them as diminishing the novelty of this submission btw).

The critical idea on which the present submission revolves is that of “standardized operating procedures” (SOPs). SOPs outline the responsibilities of each team member and establish standards for the intermediate outputs that will end up being passed between the agents.

This work innovates on the idea of message passing between agents. It uses formatted messages with specific schema per role instead of (or in addition to) arbitrary natural language.

The communication topology involves a global message pool and a subscription mechanism to prevent information overload (btw, the term “information overload” is not really explained why it should matter for LLM agents since one may think it would not, though the meaning is intuitive, and I suspect many readers will have observed the phenomenon with LLMs themselves anyway).

Benchmarks: This paper used MBPP and HumanEval. I am not familiar with the benchmarks used in the code generation field so I cannot comment on their appropriateness.

The secondary study of role ablation is interesting, though it’s presumably not a very generic result. Presumably you would get different results with a different set of coding challenges or with a different task and a different SOP. It’s still nice to see that you can do this kind of ablation here though. In practice this kind of role ablation will likely end up being an important debugging and validation step to take whenever you apply this approach to a new problem.

One question: how specific is this work to the specific SOP of a software company? Could you easily create a different kind of company, say a design company? Or an architecture firm? Maybe an artist’s workshop? Or an academic research lab? How do you actually specify an SOP? Is it essentially the config file for the entire experiment? Or is it backed in more deeply in code perhaps? Do you have to write code to change the SOP?

**Flag For Ethics Review:** No ethics review needed.

**Rating: 8:** accept, good paper

**Confidence:** 2: You are willing to defend your assessment, but it is quite likely that you did not understand the central parts of the submission or that you are unfamiliar with some pieces of related work. Math/other details were not carefully checked.

**Code Of Conduct:** Yes

---

> ### Author Response · Authors · 2023-11-19
> **Thanks for your review! Authors' feedback.**
>
> We appreciate your responsible attitude (i.e., **This note is from the area chair who is also acting as a reviewer for this paper since we didn't end up getting three reviews out of the usual process.**) and valuable time. Additionally, your positive feedback (i.e., **This is a really nice idea whose time has clearly come. I’ve noticed several versions of the basic idea appear over the last few months (all so recent that it would not be fair to consider them as diminishing the novelty of this submission btw).**) motivates us to move in a better direction.
>
> ---
>
> **Q1. This paper used MBPP and HumanEval. I am not familiar with the benchmarks used in the code generation field so I cannot comment on their appropriateness.**
>
> HumanEval and MBPP are two commonly used and popular benchmarks for code generation. They are widely utilized in related works [1,2,3].
>
> [1] OpenAI. "GPT-4 Technical Report". arXiv:2303.08774, 2023.
>
> [2] Luo, Ziyang, et al. "WizardCoder: Empowering Code Large Language Models with Evol-Instruct". arXiv:2306.08568, 2023.
>
> [3] Chen M, Tworek J, Jun H, et al. "Evaluating large language models trained on code". arXiv:2107.03374, 2021.
>
> ---
>
> **Q2. The communication topology involves a global message pool and a subscription mechanism to prevent information overload (btw, the term “information overload” s not really explain why it should matter for LLM agents since one may think it would not, though the meaning is intuitive, and I suspect many readers will have observed the phenomenon with LLMs themselves anyway).**
>
> We agree with your opinion that whether 'information overload' becomes an issue depends on the specific applications. In MetaGPT, **information overload refers to situations where agents receive more information than necessary**, for example, irrelevant messages. MetaGPT adopts the message pool to streamline one-to-one communication (refer to Sec 3.2, Message Pool), while the subscription mechanism (see Sec 3.2, Subscription Mechanism) can enhance the efficiency of context utilization by automatically filtering relevant contexts.  **In our software design scenarios and the corresponding SOPs, there is a need for more efficient and valuable communication. That's why MetaGPT needs to take 'information overload' into consideration.**
>
> ---
>
> **Q3. One question: how specific is this work to the specific SOP of a software company? Could you easily create a different kind of company, say a design company? Or an architecture firm? Maybe an artist’s workshop? Or an academic research lab? How do you actually specify an SOP? Is it essentially the config file for the entire experiment? Or is it backed in more deeply in code perhaps? Do you have to write code to change the SOP?**
>
> It's an interesting question. The MetaGPT framework, adaptable for various industries, allows users to build new agent teams by modifying code. As detailed in Sec. 3.1 and Sec. 3.2, this involves three steps:
> * 1) Configuring profiles, action functions, and tools for new agents.
> * 2) Defining each agent's structured interface, including data types and formats.
> * 3) Establishing the team's workflow and action sequence.
>
> After the initial setup, MetaGPT enables creating specialized teams without further code modifications.
>
> ---
>
> **We appreciate your time in reading our feedback and look forward to further discussion and your suggestions.**

---

### Author Response · Authors · 2023-11-22
**Global Response**

Dear all reviewers,

**Thank you again for your constructive comments and suggestions. We have accordingly improved the illustrations and added experiments to the manuscript:**

---

- We added an "A" to our title and included "LLM-based" in "Most of the current LLM-based multi-agent frameworks..." to clarify that MetaGPT is one (and one kind of) multi-agent framework. This will avoid potential misleading claims (**itRX**).
- We included more illustrations about consistency in the introduction/related work, and illustrations about beneficial interaction in related work. Besides, we cited more references to support these claims (**WkUQ**).
- We added several clarifications about the experiment setting (evaluation metrics and baselines) in 4.1 (**itRX**).
- We included results of MetaGPT with different LLMs (i.e., GPT-3.5, DeepSeek Coder) in Appendix B.2 (**uj5i**).
- We also added additional experiment about the initial input of different levels in Appendix B.2. (**itRX**)
- We included limitations and ethics concerns in Appendix C (**WkUQ and itRX**).
- We added open discussions about "deep-seated challenges" and "information overload" in the multi-agent framework in Appendix D. (**u7nd and WkUQ**).

----
**We hope these updates address the reviewers' concerns. We remain open to further discussions and revisions.**

---

### Meta-Review · Area_Chair_u7nd · 2023-12-03

**Metareview:**

This paper was very well-received by three out of the four reviewers, who all gave it a score of 8 (Note: I was one of these reviewers, see official comments). The fourth reviewer gave it a score of 3. However, it was entirely concerned with generalities that affect the entire LLM field of research, not specifics of the present paper. This is especially apparent from their spurious referral of the paper for ethics review, but also true of the rest of the review too. For instance, it nitpicks on meanings of words like "efficient" and "well-organized" that don't change the meaning of the paper. Moreover, the review does not show an understanding of the framework's novelty as a multi-agent collaboration framework where agents talk to each other in order to work together to complete a task.

**Justification For Why Not Higher Score:**

N/A

**Justification For Why Not Lower Score:**

Note that this paper received 4 reviews, one of which was mine, so the system did not count it as a "official review". The reason I reviewed the paper was that it had still only received two reviews by the deadline.  After I submitted my review (and several days into the author response period) one more of the originally assigned reviewers finally chimed in. This last reviewer gave the paper an extremely low score, and spuriously referred it for ethics review. I am of the opinion that we should ignore the low-scoring review for the reasons I explained in the metareview above. The ethics referral in particular shows bias against research on generative AI.

So after discounting the low-scoring review, all three remaining reviews for this paper all gave it a score of 8.

Note also what I commented in my review on novelty: several other similar papers and codebases have appeared over the last few months. However, all were so recent that we should not consider them as being too similar from the perspective of judging novelty. Rather we should instead consider them to be simultaneous work. The fact that so many researchers are thinking in a similar direction right now is testament to the fact that this topic of multi-agent collaboration with LLM-enabled agents is "in the air" right now, and rapidly growing in importance.

---

### Decision · Program_Chairs · 2024-01-16

Accept (oral)